# Simultaneous $Zn^{2+}$ tracking in multiple organelles using super-resolution morphology-correlated organelle identification in living cells

Hongbao Fang[1,2,3,10], Shanshan Geng[1,10], Mingang Hao[2], Qixin Chen[2], Minglun Liu[1], Chunyan Liu[4], Zhiqi Tian[2], Chengjun Wang[5], Takanori Takebe [4,6,7,8,9], Jun-Lin Guan[2], Yuncong Chen [1,3✉], Zijian Guo [1,3], Weijiang He [1,3✉] & Jiajie Diao [2✉]

$Zn^{2+}$ plays important roles in metabolism and signaling regulation. Subcellular $Zn^{2+}$ compartmentalization is essential for organelle functions and cell biology, but there is currently no method to determine $Zn^{2+}$ signaling relationships among more than two different organelles with one probe. Here, we report simultaneous $Zn^{2+}$ tracking in multiple organelles (Zn-STIMO), a method that uses structured illumination microscopy (SIM) and a single $Zn^{2+}$ fluorescent probe, allowing super-resolution morphology-correlated organelle identification in living cells. To guarantee SIM imaging quality for organelle identification, we develop a new turn-on $Zn^{2+}$ fluorescent probe, NapBu-BPEA, by regulating the lipophilicity of naphthalimide-derived $Zn^{2+}$ probes to make it accumulate in multiple organelles except the nucleus. Zn-STIMO with this probe shows that CCCP-induced mitophagy in HeLa cells is associated with labile $Zn^{2+}$ enhancement. Therefore, direct organelle identification supported by SIM imaging makes Zn-STIMO a reliable method to determine labile $Zn^{2+}$ dynamics in various organelles with one probe. Finally, SIM imaging of pluripotent stem cell-derived organoids with NapBu-BPEA demonstrates the potential of super-resolution morphology-correlated organelle identification to track biospecies and events in specific organelles within organoids.

[1] State Key Laboratory of Coordination Chemistry, School of Chemistry and Chemical Engineering, Nanjing University, 210023 Nanjing, China. [2] Department of Cancer Biology, University of Cincinnati College of Medicine, Cincinnati, OH 45267, USA. [3] Chemistry and Biomedicine Innovation Center, Nanjing University, 210023 Nanjing, China. [4] Division of Gastroenterology, Hepatology and Nutrition, Cincinnati Children's Hospital Medical Center, Cincinnati, OH 45267, USA. [5] Sinopec Shengli Petroleum Engineering Limited Company, Dongying, China. [6] Center for Stem Cell and Organoid Medicine (CuSTOM), Cincinnati Children's Hospital Medical Center, Cincinnati, OH 45267, USA. [7] Division of Developmental Biology, Cincinnati Children's Hospital Medical Center, Cincinnati, OH 45267, USA. [8] Department of Pediatrics, University of Cincinnati College of Medicine, Cincinnati, OH 45267, USA. [9] Institute of Research, Tokyo Medical and Dental University, 1-5-45 Yushima, Bunkyo-ku, Tokyo, Japan. [10] The authors contributed equally: Hongbao Fang, Shanshan Geng. ✉email: chenyc@nju.edu.cn; heweij69@nju.edu.cn; jiajie.diao@uc.edu

Zinc homeostasis, including labile $Zn^{2+}$ fluctuation, is closely associated with many physiological processes and disease pathologies[1–3]. As the newly proposed "second messenger", labile $Zn^{2+}$ is involved in both inter- and intracellular signal regulation and transmission[4–7], and new phenomena for labile $Zn^{2+}$-associated signal transduction, such as *zinc sparks* and *zinc waves*, have been observed using fluorescence imaging[8,9]. Except for various organelles having different $Zn^{2+}$ levels and dynamics[10], many physiological processes involving multiple organelles are also associated with $Zn^{2+}$ fluctuation. In particular, a growing body of evidence suggests that $Zn^{2+}$ was pivotal to autophagy and autophagy could prompt significant changes in intracellular $Zn^{2+}$[11–15]. As a dynamic process for cells to degrade and recycle proteins and senescent organelles to overcome stressful conditions, autophagy is associated with aging, neurodegenerative diseases, diabetes, and fatty liver disease, and is a potential target for diagnosis and treatment[16–18]. It is therefore important to determine $Zn^{2+}$ signaling relationships among organelles during autophagy and other cellular processes. Towards this goal, we developed the concept of simultaneously tracking $Zn^{2+}$ in multiple organelles (Zn-STIMO), which is both appealing and challenging.

Synchrotron X-ray radiation fluorescence microscopy (SXRF) has been used for zinc mapping[8,19,20], and its correlation with transmission electron microscopy (TEM) might provide zinc information in multiple organelles simultaneously in dehydrated cells without differentiating labile and closely associated $Zn^{2+}$. On the other hand, fluorescence imaging would have the advantage of providing dynamic labile $Zn^{2+}$ information in living cells. An indirect strategy of multi-round $Zn^{2+}$ fluorescence confocal imaging via double-staining with either two different organelle-targeting $Zn^{2+}$ probes or one multiple-organelle-distributed $Zn^{2+}$ probe and one organelle-targeting dye has been developed to track labile $Zn^{2+}$ in different organelles[21]. However, imaging labile $Zn^{2+}$ dynamics in multiple organelles in a single procedure would be a more reliable way to unravel the $Zn^{2+}$ signaling pathways among these organelles. A single fluorescence imaging procedure via multi-staining with a series of $Zn^{2+}$ probes targeting different organelles is a logical idea, yet development of a series of $Zn^{2+}$ probes without signal leakage is chemically challenging. Fluorescence imaging via staining with only one $Zn^{2+}$ probe distributing in multiple organelles is another alternative, yet it would require the spatial resolution to identify organelles in living cells based on their morphological features, similar to that of TEM for dehydrated fixed cells.

With the maturation of super-resolution fluorescence microscopic techniques[22–24], morphology-correlated organelle identification could be achieved for most organelles. Therefore, fluorescence microscopic techniques acquiring super-resolution may be ideal for simultaneously tracking $Zn^{2+}$ in multiple organelles[25,26], and a synthetic fluorophore that distributes to multiple organelles is demanded. Considering the size of normal organelles such as mitochondria (500–5000 nm), endoplasmic reticulum (ER; size not given due to their unique morphology), lysosomes (100–1200 nm), nucleus (~10 μm), and autophagosomes (150–1500 nm)[27], structured illumination microscopy (SIM), with limited spatial resolution of ~100 nm[28–30], and good compatibility with conventional synthetic fluorophores, might be a promising alternative. Indeed, our previous SIM study using synthetic probes enabled us to visualize organelle interactions via morphology-correlated organelle identification[29–33].

Here, we present Zn-STIMO using a single $Zn^{2+}$ fluorescent probe and SIM imaging for morphology-correlated organelle identification (Fig. 1a). Using the technique to investigate HeLa cells undergoing autophagy revealed different labile $Zn^{2+}$ enhancement factors in the mitochondria and ER, with the highest labile $Zn^{2+}$ appearing in the autophagosomes/autolysosomes. SIM imaging of organoids displayed the potential of super-resolution morphology-correlated organelle identification to track biospecies and events in specific organelles within organoids.

## Results

**Designing a candidate $Zn^{2+}$ probe for Zn-STIMO**. We considered the properties that a $Zn^{2+}$ fluorescent probe would require for Zn-STIMO: the probe should (1) be photostable to reduce interference from probe photobleaching, especially for a turn-on probe, (2) have high fluorescence quantum yield to reduce imaging background[34], (3) show a reversible $Zn^{2+}$-specific response, (4) have low cytotoxicity, and (5) distribute to multiple organelles.

Fluorophore 1,8-naphthalimide (Nap) with a large aromatic structure, low probability of non-radiative transition, and high fluorescence quantum yield[35,36], would be a promising parent fluorophore to construct a $Zn^{2+}$ probe for Zn-STIMO. Naph-BPEA, a Nap-derived $Zn^{2+}$ probe formed by integrating $Zn^{2+}$ chelator BPEA (N,N′-bis(pyridin-2-ylmethyl)ethane-1,2-diamine) at Nap's 4-position[37,38], accumulates in multiple organelles showing punctate fluorescence in the cytoplasm. However, exogenous $Zn^{2+}$ loading causes this probe to redistribute uniformly inside the cell including the nucleus[39], making it unsuitable for organelle identification and Zn-STIMO in living cells. The subcellular distribution of fluorophores, especially those without targeting group, can be altered by changing their lipophilicity. Therefore, replacing the ethylglycol tail with alkyl chains, such as ethyl and butyl groups, creating NapEt-BPEA and NapBu-BPEA, respectively, might enhance the probe's lipophilicity to change the probe's intracellular accumulation behavior (Fig. 1b). The calculation of logP (partition coefficient of free probe in n-octanol/water) and logP$_{Zn}$ (partition coefficient of probe's zinc complex in n-octanol/water) for probes Naph-BPEA, NapEt-BPEA and NapBu-BPEA disclosed that the NapBu-BPEA possessed the highest logP and logP$_{Zn}$ values among the three, we hypothesized that this probe has the desired subcellular distribution behavior, although NapEt-BPEA shows the similar accumulation behavior to that of Naph-BPEA. Theoretical calculation of Naph-BPEA and NapBu-BPEA revealed that all their frontier molecule orbitals at the same state are similar in electron densities and energies (Fig. 1c and Supplementary Table 2), suggesting their different tails, ethylglycol and butyl groups, have a limited effect on their frontier molecule orbitals. The energy gaps between the first singlet excited state S1 and the first triplet excited state T1, $\Delta E_{ST}$, for the two probes were ~1.04 eV (Fig. 1c). This large energy gap indicated that these Nap/BPEA-derived probes are unfavorable for intersystem crossing (ISC) to form reactive oxygen species (ROS), benefiting probe's photostability and cell viability by averting ROS-induced damage. However, the contribution of the intrinsic structural stability of rigid Nap should not be excluded. Overall, we expect NapBu-BPEA to be photostable, which is essential for labile $Zn^{2+}$ tracking to avoid interference from photobleaching, especially when the excitation power for SIM imaging is higher than that for confocal imaging.

**NapBu-BPEA shows reversible $Zn^{2+}$-specific sensing behavior**. We determined the spectroscopic sensing behavior of NapBu-BPEA for $Zn^{2+}$ in HEPES buffer. NapBu-BPEA exhibited a broad absorption band with a maximum at 454 nm ($\varepsilon$, $1.226 \times 10^4\,M^{-1}cm^{-1}$) in its absorption spectrum (Supplementary Fig. 7), and a typical emission band of Naps centered at 540 nm in its emission spectrum. Its emission quantum yield was 0.32 ($\lambda_{ex}$, 454 nm, Supplementary Table 1), which is higher than that of Naph-BPEA[39].

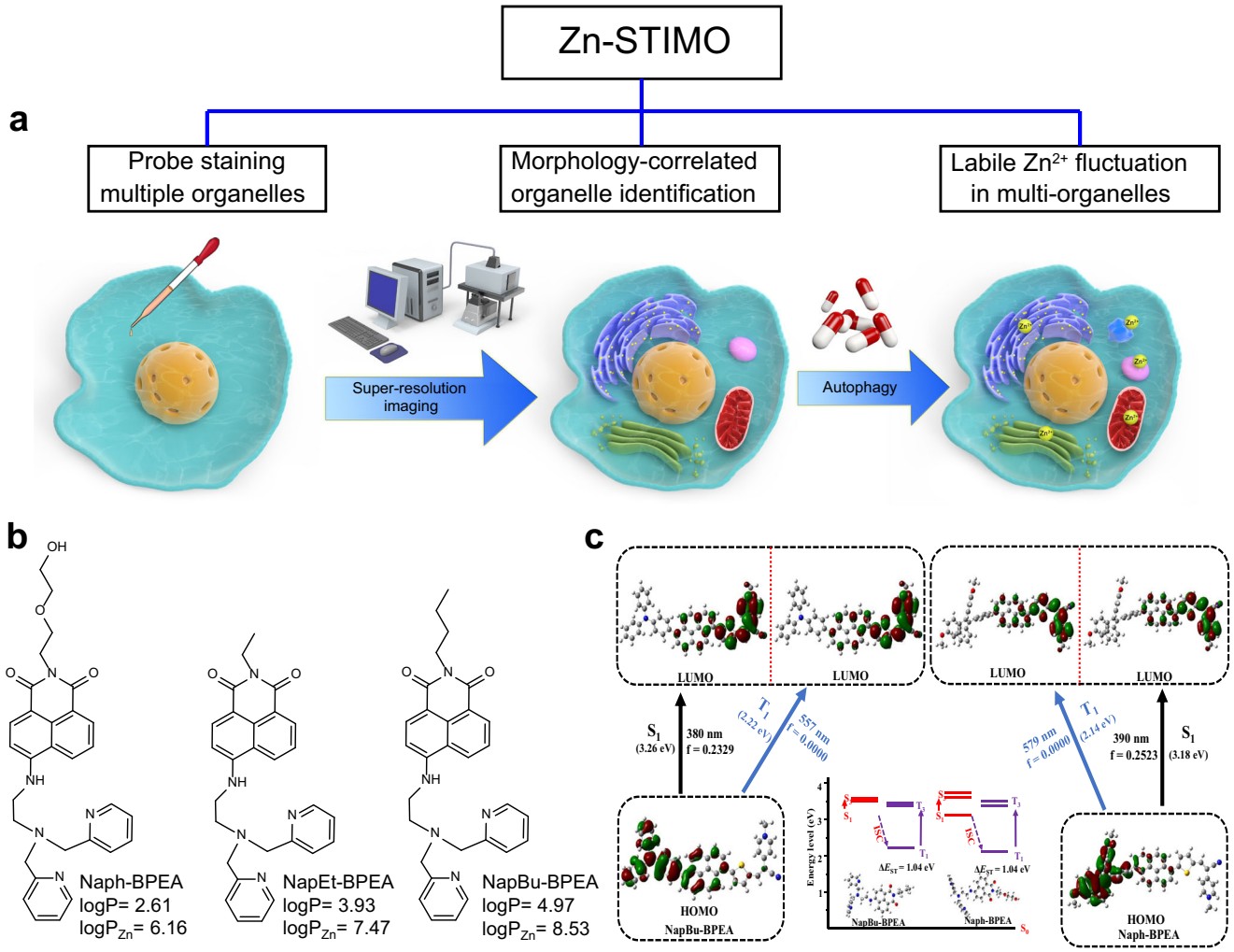

**Fig. 1 Schematic illustration of Zn-STIMO and the design of probe candidate for Zn-STIMO. a** General scheme of Zn-STIMO; **b** fluorescent $Zn^{2+}$ probes Naph-BPEA, NapEt-BPEA, and NapBu-BPEA. LogP values are for free probes, while $logP_{Zn}$ values are for their zinc complexes formed with $ZnCl_2$; **c** calculated frontier molecule orbitals and energies of NapBu-BPEA and Naph-BPEA. Calculation was performed using TD-DFT//B3LYP/6-31G(d) level based on the geometry optimized by the DFT//B3LYP/6-31G(d) basis set.

Fluorescent $Zn^{2+}$ titration of NapBu-BPEA revealed a linear enhancement of emission (Fig. 2a and Supplementary Fig. 8a) and a slight emission blue shift to 535 nm. The fluorescence intensity became stable when 1 eq $Zn^{2+}$ was added, suggesting a $Zn^{2+}$ binding stoichiometry of 1:1. This $Zn^{2+}$-induced fluorescence enhancement can be attributed to the blockage of the photoinduced electron transfer (PET) effect triggered by BPEA chelating $Zn^{2+}$[38,40–43]. The emission enhancement factor at 540 nm, $F/F_0$, attained to 3.6-fold, and the emission quantum yield upon $Zn^{2+}$ binding was enhanced to 0.80. Fluorescence job's plot confirmed the 1:1 $Zn^{2+}$ binding stoichiometry of this probe (Supplementary Fig. 8b). We also verified this binding mode through high-resolution mass spectrometric determination of NapBu-BPEA solution added with 1 eq $ZnCl_2$, which revealed a $m/z$ ratio of 592.1523 assigned as $[NapBu-BPEA + ZnCl]^+$ (Supplementary Fig. 9). Absorption titration of NapBu-BPEA with $Zn^{2+}$ led to a minor hypsochromic shift to 445 nm ($\varepsilon$, $1.044 \times 10^4 M^{-1}cm^{-1}$) and a slight absorbance decrease. The absorption spectra became stable after the addition of 1 eq of $Zn^{2+}$ (Supplementary Fig. 10b), which is consistent with the fluorescence titration results.

We investigated the $Zn^{2+}$-specific sensing selectivity of NapBu-BPEA by screening metal cations of interest for their

fluorescent response (Fig. 2b). NapBu-BPEA showed no obvious fluorescence change in the presence of cell abundant metal cations such as $Na^+$, $K^+$, $Ca^{2+}$, and $Mg^{2+}$ (1000 eq), or other cations such as $Cr^{3+}$, $Pb^{2+}$, $Al^{3+}$, $Fe^{2+}$, $Fe^{3+}$, $Ba^{2+}$, and $Mn^{2+}$ (1 eq). Besides the instant $Zn^{2+}$-induced fluorescence enhancement (3.6-fold), this probe showed slight enhancement (1.5-fold) induced by $Cd^{2+}$ (1 eq), and minor fluorescence decrease induced by $Co^{2+}$ and $Ni^{2+}$ (1 eq). The other metal cations did not interfere with the probe's $Zn^{2+}$-specific turn-on response, except for $Co^{2+}$ and $Ni^{2+}$, which reduced the $Zn^{2+}$-induced enhancement. Due to the scarcity of $Cd^{2+}$, $Co^{2+}$, and $Ni^{2+}$ in cells, these interferences were negligible for $Zn^{2+}$ imaging in living cells. The $Zn^{2+}$-enhanced fluorescence of NapBu-BPEA could be recovered back to that of free probe by a metal chelator $N,N,N',N'$-tetrakis-2-pyridylmethyl ethylenediamine (TPEN) to remove $Zn^{2+}$. This rapid reversible $Zn^{2+}$ sensing behavior can be repeated for many cycles without obvious attenuation (Fig. 2c and Supplementary Fig. 8c). The dissociation constant ($K_d$) of the $Zn^{2+}$/NapBu–BPEA complex was 4.98 nM (Fig. 2d and Supplementary Fig. 8d). NapBu-BPEA showed a $Zn^{2+}$ $K_d$ value locating exactly in the range from 0.2 to 70 nM (Supplementary Table 3), indicating its $Zn^{2+}$ affinity is suitable for sensing labile $Zn^{2+}$ in organelles just as these reported probes[36,44,45]. As the $Zn^{2+}$ buffer

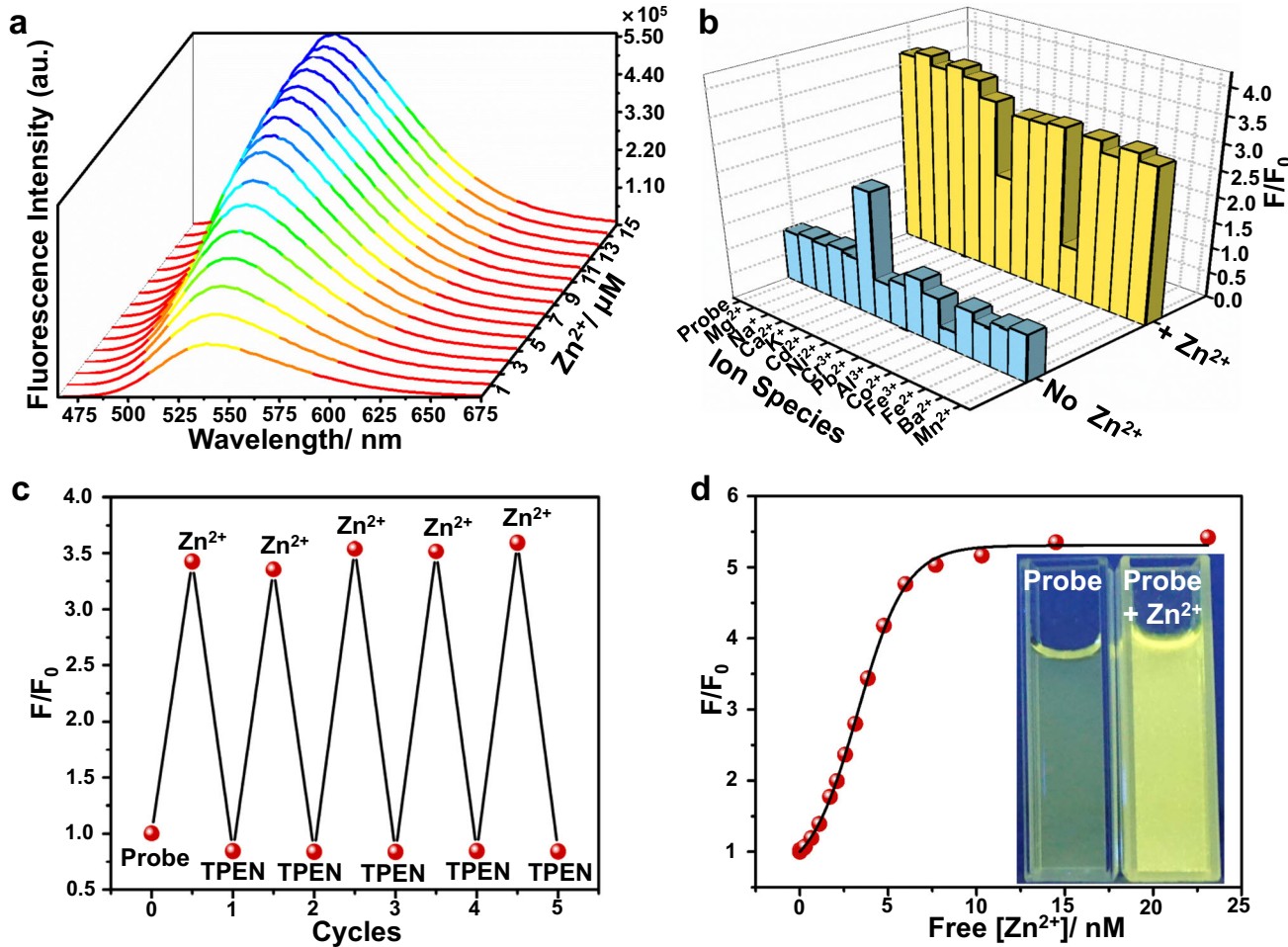

**Fig. 2 NapBu-BPEA shows reversible $Zn^{2+}$-specific sensing behavior.** Fluorescence spectroscopic determination of NapBu-BPEA (10 μM) in HEPES buffer upon excitation at 450 nm. **a** Emission spectra determined upon $Zn^{2+}$ titration (10 mM, 0.3 μL aliquot); **b** fluorescence enhancement factor of NapBu-BPEA induced by $K^+$, $Na^+$, $Ca^{2+}$, $Mg^{2+}$ (1000 eq), $Cd^{2+}$, $Ni^{2+}$, $Cr^{3+}$, $Pb^{2+}$, $Al^{3+}$, $Co^{2+}$, $Fe^{2+}$, $Fe^{3+}$, $Ba^{2+}$, and $Mn^{2+}$ (1 eq), and the $Zn^{2+}$ (1 eq) induced enhancement factor in the presence of above cations. All are based on emission at 540 nm; **c** fluorescent response of NapBu-BPEA based on emission at 540 nm in cycles of $Zn^{2+}$ (10 μM) addition and subsequent TPEN (10 μM) treatment; **d** fluorescence of NapBu-BPEA at 540 nm determined in HEPES buffers containing different $[Zn^{2+}]_{total}$ for binding constant calculation. Inset in **d**: photograph of NapBu-BPEA solutions before and after $Zn^{2+}$ addition (1 eq) under UV lamp of 365 nm.

proteins, metallothioneins (MTs), are labile zinc source in cells. They normally bind with $Zn^{2+}$ showing a $K_d$ value of 0.32 pM at pH = 7.4. Therefore, probe NapBu-BPEA is difficult to take $Zn^{2+}$ from MTs. However, this probe is able to sense labile $Zn^{2+}$ release from MTs induced by various stresses and stimulations[46]. Fluorescence determination at different pH values (from pH 4.0–8.0) showed that this probe's $Zn^{2+}$-specific turn-on sensing behavior was pH-independent (Supplementary Fig. 11). Therefore, NapBu-BPEA seems a promising candidate for intracellular $Zn^{2+}$ fluorescence imaging.

**NapBu–BPEA distributes to multiple organelles in HeLa cells.** We determined the cytotoxicity of the probe by measuring the activity of lactate dehydrogenase released into the medium using a water-soluble tetrazolium (WST) kit. The cell viability after 24 h of incubation with 20 μM probe was above 80% (Supplementary Fig. 12), indicating that NapBu–BPEA has low cytotoxicity and is suitable for cell imaging. We then investigated the intracellular imaging ability of NapBu-BPEA in HeLa cells using SIM. Punctate fluorescence signal appeared in the cytoplasm but not in the nucleus after 10 min of probe incubation, and the signal tended to

stabilize after 1 h of incubation (Supplementary Fig. 13e), confirming the probe's cell membrane permeability. We subsequently performed all cell imaging by staining cells with 10 μM NapBu-BPEA for 1 h.

We performed co-localization imaging of NapBu-BPEA with LysoTracker Red (LTR), MitoTracker Deep Red (MTDR), ER-Tracker Red (ERTR), and Hoechst 33258 (Hoechst) in HeLa cells. As shown in Fig. 3, the probe's Pearson's correlation coefficients (PCCs) with LTR, MTDR, ERTR, and Hoechst dye were 0.509. 0.563, 0.717, and 0.086, indicating that NapBu–BPEA accumulated effectively in all the stained organelles except the nucleus. Moreover, the insets in Fig. 3a–i suggested that NapBu-BPEA was able to visualize the typical morphologies of organelles such as the mitochondria, lysosomes, and ER. SIM imaging of HeLa cells after 1 h of NapBu-BPEA incubation clearly showed punctate lysosomes, rod mitochondria with distinct cristae, and reticular ER (Fig. 3m iv), while confocal imaging provided little information on organelle morphology (Fig. 3m iii). This indicates that SIM imaging via NapBu-BPEA staining can achieve morphology-correlated organelle identification in living cells. Fluorescence tracking on the line across the mitochondrion in Fig. 3m iv revealed that SIM imaging via NapBu-BPEA staining

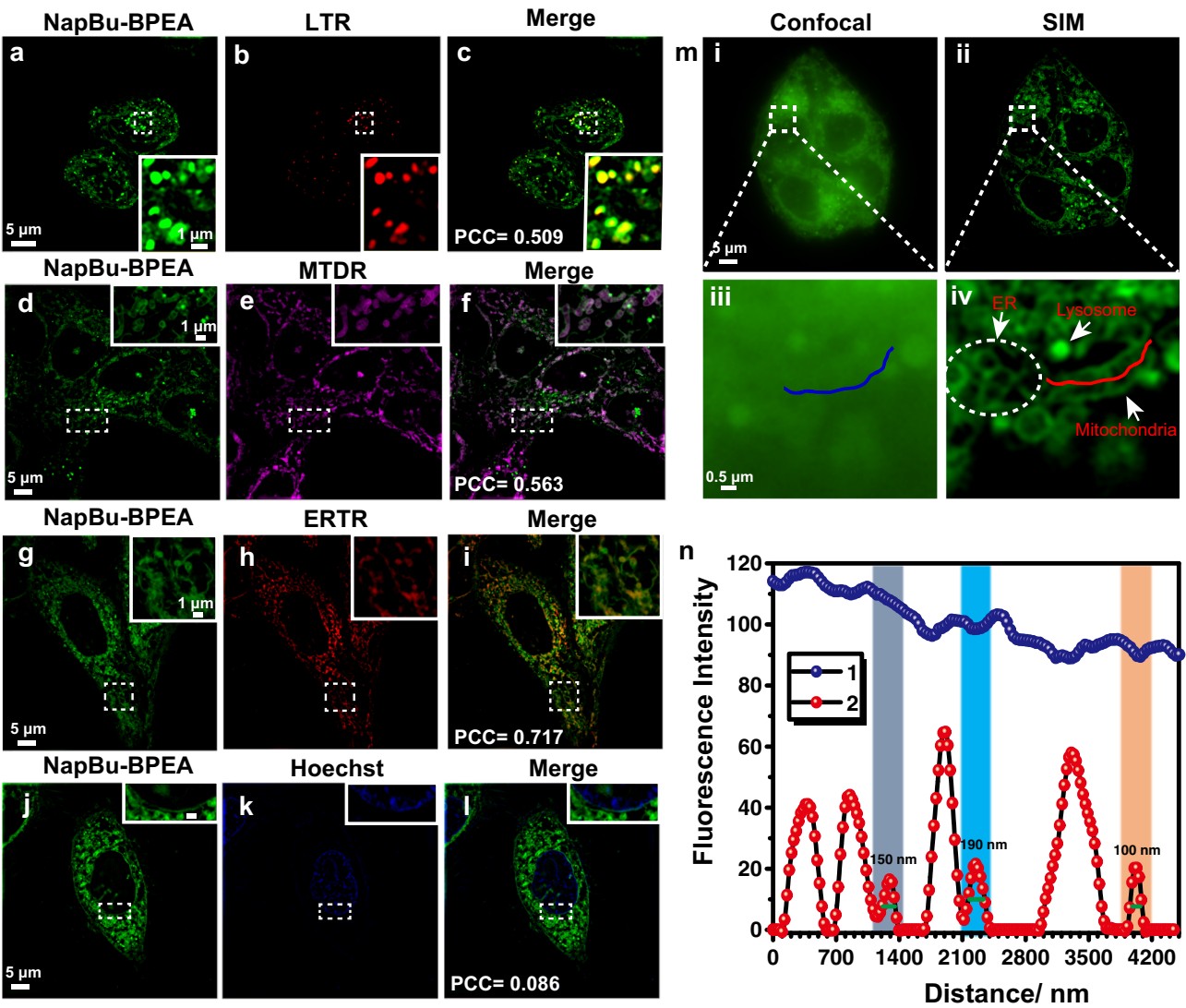

**Fig. 3 NapBu–BPEA distributes to multiple organelles in HeLa cells. a–l** SIM images for HeLa cells co-stained by NapBu–BPEA (10 μM) with **a–c** LysoTracker Red (LTR, 0.1 μM), **d–f** MitoTracker Deep Red (MTDR, 0.5 μM), **g–i** ER-Tracker Red (ERTR, 1 μM), **j–l** nuclear dye Hoechst 33258 (Hoechst, 1 μg/mL). All insets give the enlarged images for the dashed pane. **m** Confocal (i and iii) and SIM (ii and iv) images for the NapBu-BPEA-stained HeLa cells, and **n** the fluorescence signal detected on the lines across a specific mitochondrion in (iii) and (iv). Scale bar: 5 μm, enlarged images scale bar: 0.5 μm.

was able to distinguish the signal with a full width at half maximum (FWHM) down to 100 nm (Fig. 3n). This resolution was similar to that achieved with the mitochondrial dye Atto 647 N (FWHM, 91 nm) by Han and coworkers[47], confirming this $Zn^{2+}$ probe's potential for Zn-STIMO in living cells via morphology-correlated organelle identification.

We also investigated the intracellular photostability of NapBu–BPEA in SIM imaging in a long-term (300 s) excitation process for the cells co-stained by NapBu–BPEA and MTDR (Supplementary Fig. 14). The distinct intracellular fluorescence of NapBu–BPEA was almost stable during the excitation process, while the MTDR fluorescence decreased clearly in the same process, and the fluorescence intensity was <30% of the initial signal after 120 s of irradiation. The higher photostability of this probe than MTDR suggested that the intracellular fluorescence signal of NapBu–BPEA would enable long-term super-resolution $Zn^{2+}$ dynamic imaging without distinct interference from photobleaching.

**NapBu-BPEA can sense intracellular labile $Zn^{2+}$ in a reversible manner.** We next investigated NapBu–BPEA for its intracellular

labile $Zn^{2+}$ imaging ability. We performed the imaging experiments in HeLa cells upon exogenous $Zn^{2+}$ loading via incubation with zinc pyrithione cmplex (ZnPT), a cell-permeable $Zn^{2+}$ carrier[44]. We tracked the intracellular labile $Zn^{2+}$ enhancement processes by recording SIM images every 2 min. The imaging revealed an instant fluorescence enhancement upon ZnPT (50 μM) incubation, displaying an almost linear increase in the first 4 min (Fig. 4a, b). The stable fluorescence signal after 4 min indicated that the exogenous $Zn^{2+}$ loading process is completed quickly. During the $Zn^{2+}$ loading process, in addition to the intensity enhancement, the punctate fluorescence in the cytoplasm was retained and no fluorescence appeared in the nucleus even after 1 h of ZnPT incubation. This intracellular distribution behavior was clearly different from the redistribution behavior of Naph-BPEA upon $Zn^{2+}$ loading, which resulted in the uniform fluorescence in the cytoplasm and nucleus. We propose that the butyl tail of NapBu-BPEA is responsible for the retained multiple organelle accumulation behavior of NapBu-BPEA upon $Zn^{2+}$ binding.

Higher ZnPT concentration led to higher fluorescence enhancement in the cytoplasm (Fig. 4c–h), implying that NapBu-BPEA enables fluorescence tracking of labile $Zn^{2+}$ enhancement in cells.

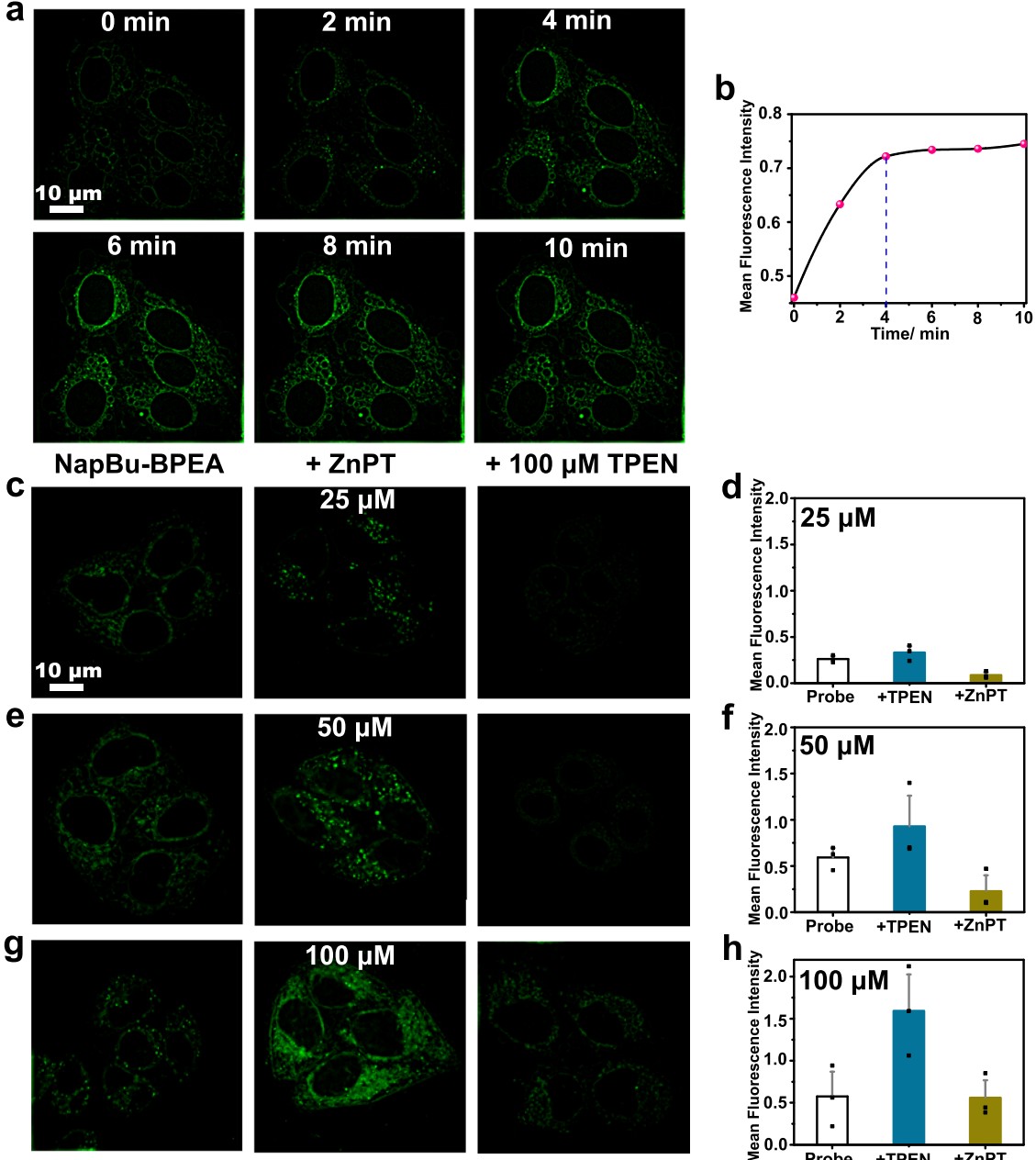

**Fig. 4 NapBu-BPEA can sense intracellular labile Zn²⁺ in a reversible manner. a** SIM imaging of the NapBu-BPEA-stained HeLa cells (10 μM, 1 h, 37 °C) upon incubation with 50 μM ZnPT; and **b** the corresponding temporal profile of intracellular fluorescence increase. **c–h** SIM imaging of the probe-stained (10 μM, 1 h, 37 °C) HeLa cells undergoing ZnPT (25 μM, **c**, **d**; 50 μM, **e**, **f**; 100 μM, **g**, **h**) incubation (10 min) and subsequent TPEN (100 μM, 30 min) treatment; and the corresponding histograms (**d**, **f**, **h**) of the determined average fluorescence intensity in cells from $n = 3$ biologically independent experiments, mean ± SD. Scale bar, 10 μm. Source data are available as a Source Data file.

Intracellular $Zn^{2+}$ scavenging for the $Zn^{2+}$-loaded HeLa cells with TPEN (a cell membrane permeable $Zn^{2+}$ scavenger) treatment resulted in a distinct drop in cytoplasmic fluorescence. This confirmed that the fluorescence enhancement upon ZnPT incubation was really associated with labile $Zn^{2+}$ enhancement, and NapBu-BPEA was able to sense intracellular labile $Zn^{2+}$ in a reversible manner. The punctate fluorescence distribution patten was still retained in this $Zn^{2+}$ scavenging process.

**NapBu-BPEA reveals labile $Zn^{2+}$ dynamics in autophagic HeLa cells.** With the reversible $Zn^{2+}$ imaging ability and the multiple organelle distribution behavior upon $Zn^{2+}$ binding, we further investigated NapBu-BPEA for its application in Zn-

STIMO for autophagic cells. We used HeLa cells incubated with carbonyl cyanide *m*-chlorophenylhydrazone (CCCP, a mitochondrial damage inducer) as a mitophagy model. We co-stained the cells with NapBu-BPEA and DAPRed, an autophagy dye incorporating into the autophagosome during double-membrane formation via structural features and emitting under hydrophobic conditions[48,49]. As shown in Fig. 5 and Supplementary Fig. 15a–f, SIM imaging in the DAPRed channel for HeLa cells without CCCP incubation showed no red fluorescence, while the red fluorescence in the cells with 24 h of CCCP exposure (10 μM) confirmed the mitophagy induction. SIM imaging in the NapBu-BPEA channel showed that the green fluorescence intensity of NapBu-BPEA in the mitophagic cells was ~2.0-fold higher than

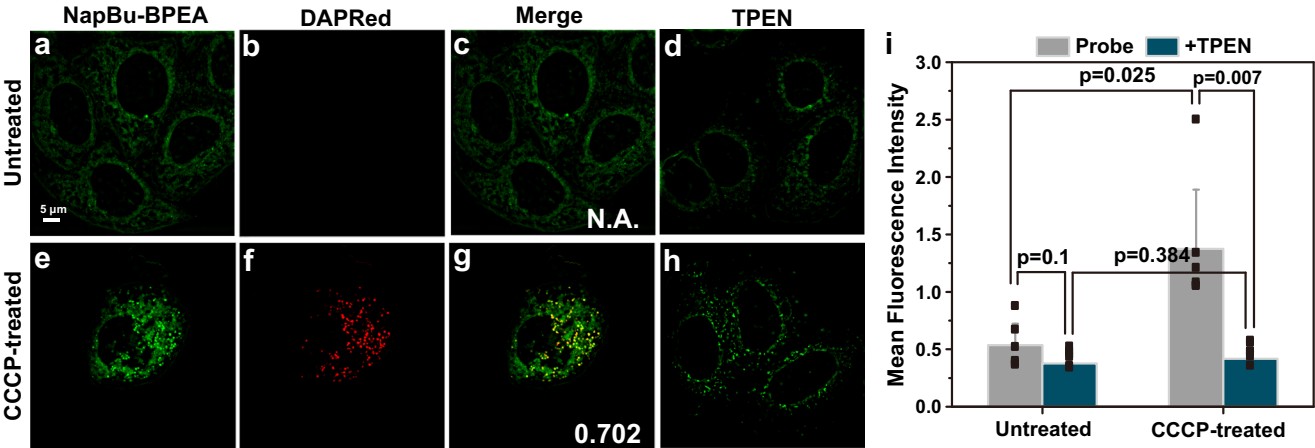

**Fig. 5 NapBu-BPEA reveals labile Zn$^{2+}$ dynamics in autophagic HeLa cells.** SIM images of HeLa cells stained by NapBu–BPEA and DAPRed without (**a–d**) or with (**e–h**) CCCP (10 μM, 24 h, 37 °C) treatment. **a, e** cell images from the green channel for NapBu–BPEA (10 μM, 1 h, 37 °C) fluorescence; **b, f** cell images from the red channel for DAPRed (1 μM, 30 min, 37 °C) fluorescence; **c, g** merged images of red and green channel images; (**d, h**) cell images for cells in **a, b** underwent TPEN (100 μM, 30 min, 37 °C) treatment; **i** average fluorescence intensity recorded in the green channel images for cells with or without CCCP treatment. $n = 6$ biologically independent experiments per group, mean ± SD, the statistical differences between the experimental groups were analyzed by double-tailed Student's $t$ test, when $p < 0.05$, it was considered to have statistical significance. Scale bar, 5 μm. Source data are available as a Source Data file.

that in non-autophagic cells (Fig. 5i and Supplementary Fig. 15j). Subsequent TPEN treatment decreased the fluorescence to a level slightly lower than that of cells without autophagy. This reversible fluorescence response in the green channel indicated that the detected NapBu-BPEA fluorescence change was caused by labile Zn$^{2+}$ fluctuation, and that NapBu-BPEA was able to visualize labile Zn$^{2+}$ fluctuation in mitophagy. In addition, the dynamic labile Zn$^{2+}$ tracking in cells exposure to CCCP (20 μM) disclosed that the autophagy induction processes underwent rapid enhancement of labile Zn$^{2+}$ in the initial 5 min of CCCP incubation, followed by and the subsequent slower Zn$^{2+}$ enhancement (Supplementary Fig. 16), and the temporal profile of the reversible labile Zn$^{2+}$ decrease induced by TPEN treatment 14 mins post CCCP incubation was also observed.

We also incubated ATG13 KO HeLa cells and FIP200 KO HeLa cells with CCCP. These cells lines were constructed using CRISPR/Cas9 gene editing technology to knock out ATG13 and FIP200, which are essential proteins for autophagy induction[50,51]. SIM imaging in the DAPRed channel showed that CCCP incubation did not induce the red fluorescence, suggesting that ATG13 or FIP200 knockout blocked the CCCP-induced mitophagy (Supplementary Fig. 17b, e, h, k). In addition, there was no fluorescence enhancement induced by CCCP in the green channel, implying the absence of labile Zn$^{2+}$ enhancement (Supplementary Fig. 17m). These observations confirmed that the CCCP-induced labile Zn$^{2+}$ release in normal HeLa cells was really associated with the induced autophagy.

As CCCP-induced mitophagy is recognized as selective autophagy for maintaining cellular homeostasis[52], non-selective autophagy such as that induced by Eagle's Balanced Salt Solution (EBSS) is also an interesting topic in this field[53,54]. SIM imaging in the DAPRed channel showed red fluorescence after 24 h of EBSS incubation, indicating the occurrence of autophagy (Supplementary Fig. 15h). However, SIM imaging in the NapBu-BPEA channel showed only the decreased fluorescence, indicating reduced labile Zn$^{2+}$ level in non-selective autophagy (Supplementary Fig. 15g, j).

**Zn-STIMO reveals different labile Zn$^{2+}$ responses in mitochondria, autophagosomes/autolysosomes, and ER during**

**autophagy.** For the CCCP-incubated HeLa cells, the PCC of the green fluorescence with the red fluorescence was ~0.70 (Fig. 5e–g), and the merged image indicated that most of the brighter green fluorescent dots coincided with the red fluorescent dots, confirming that NapBu-BPEA accumulated also in autophagosomes and autolysosomes. We therefore used NapBu-BPEA to explore Zn$^{2+}$ simultaneously in the mitochondria, autophagosomes/autolysosomes, and ER in HeLa cells upon incubation with CCCP or EBSS (Fig. 6 and Supplementary Fig. 18). We performed confocal imaging at the same time for comparison.

Confocal images for the intracellular micro regions of a HeLa cell stained by NapBu-BPEA showed blurred fluorescence (Fig. 6b, d, j, l), while SIM images gave clear morphologies of most organelles. The mitochondria (Fig. 6e, m) and the ER (Fig. 6g, o) were easily identifiable in both normal and mitophagy-stimulated HeLa cells. The autophagosomes and autolysosomes appeared as the brighter dots in the mitophagic cells induced by CCCP incubation (Fig. 6i, k). The detected NapBu-BPEA fluorescence change in the randomly selected mitochondria in Fig. 6a, i indicated that the average mitochondrial labile Zn$^{2+}$ level in mitophagic cells was almost ~1.89 times as that in normal HeLa cells. Similarly, the average labile Zn$^{2+}$ level in the ER of mitophagic cells was ~1.55 times as that in normal HeLa cells. The highest labile Zn$^{2+}$ level appeared in the induced autophagosomes or autolysosomes (Fig. 6q).

With these identified organelles, the different labile Zn$^{2+}$ responses in autophagy can also be clarified by constructing a thermal map of Zn$^{2+}$ level via fluorescence intensity analysis. A Zn$^{2+}$ thermal map for the mitochondria demonstrated a heterogeneous distribution of labile Zn$^{2+}$, and the labile Zn$^{2+}$ level in different cristae of the same mitochondrion was variable (Fig. 6e1, m1). There was no labile Zn$^{2+}$ observed in the mitochondrial matrix, but this could be due to the absence of probe in the matrix. We observed a similar non-uniform distribution of labile Zn$^{2+}$ in the ER (Fig. 6g1, o1). The highest labile Zn$^{2+}$ levels in the autophagosomes and autolysosomes appeared as the red dots in Zn$^{2+}$ thermal maps. For the EBSS-induced autophagy, we observed decreased labile Zn$^{2+}$ in the mitochondria and ER (Supplementary Fig. 18i), with labile Zn$^{2+}$ in mitochondria being higher than that in ER.

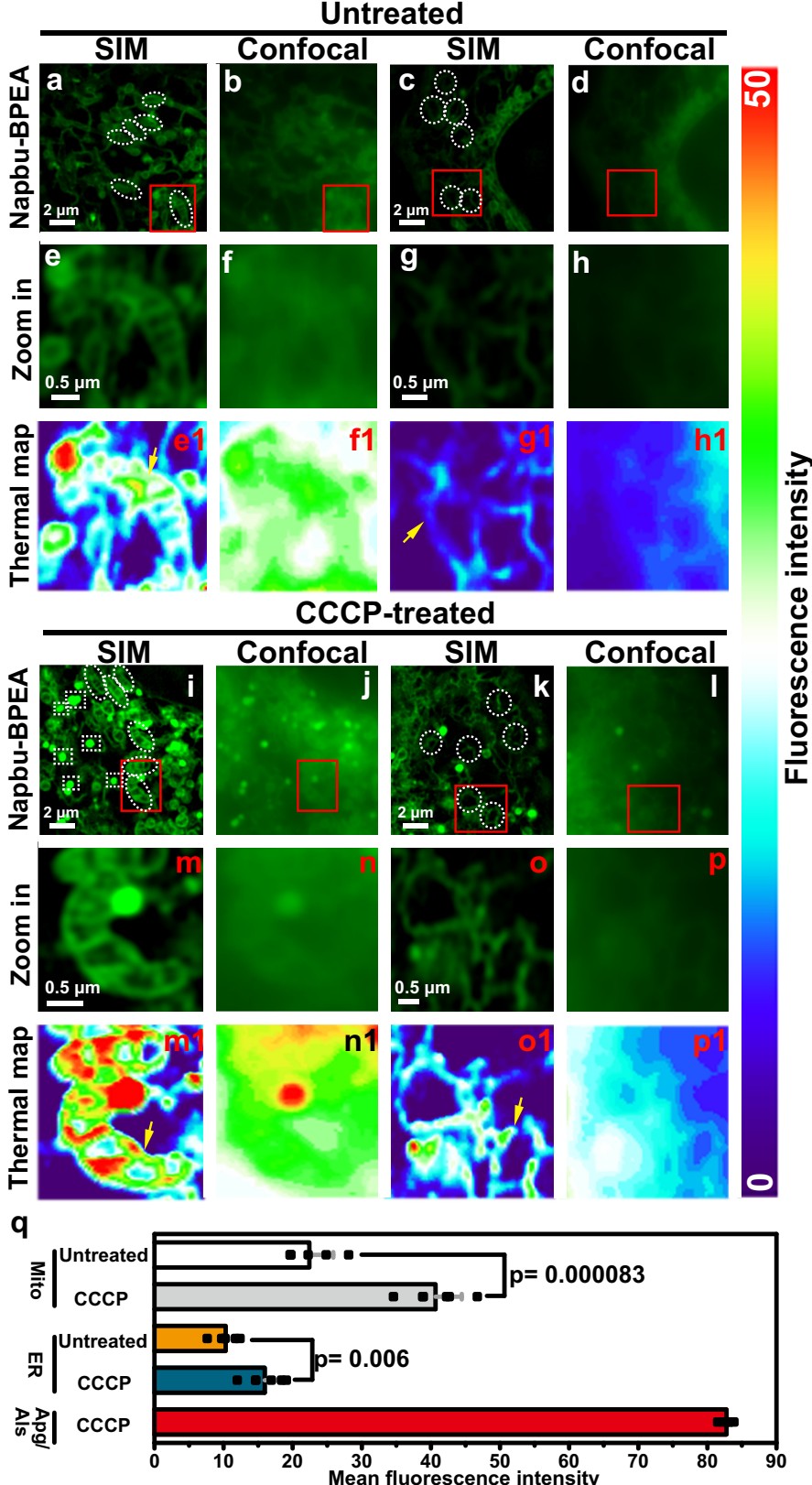

Since NapBu-BPEA can be used to monitor $Zn^{2+}$ in different organelles simultaneously via Zn-STIMO. However, the $Zn^{2+}$ level in cells changes from time to time, so tracking dynamic $Zn^{2+}$ in living cells is very attractive. In view of the previous solid experimental results, we also performed dynamic labile $Zn^{2+}$ tracking in the CCCP-induced mitophagy of HeLa cells via Zn-STIMO (Fig. 7). The temporal profiles of mean fluorescence in the mitochondria, ER, and autophagosome/autolysosome (Aps/Als) revealed that the distinct labile $Zn^{2+}$ enhancement appeared 5 min later than CCCP incubation, and the Aps/Als displayed the

**Fig. 6 Zn-STIMO reveals different labile Zn$^{2+}$ responses in mitochondria, autophagosomes/autolysosomes, and ER during autophagy.** Simultaneous Zn$^{2+}$ tracking in the mitochondria and ER of HeLa cells undergoing autophagy induced by CCCP (10 µM, 37 °C, 24 h) incubation via SIM imaging using NapBu-BPEA (10 µM, 37 °C, 1 h prior to imaging). Confocal imaging was also performed for comparison. SIM (**a**, **c**, **i**, **k**) and confocal (**b**, **d**, **j**, **l**) images of NapBu-BPEA-stained HeLa cells without CCCP treatment (SIM: **a**, **c**; confocal: **b**, **d**) or with CCCP treatment (SIM: **i**, **k**; confocal: **j**, **l**); **e–h**, **m–p** enlarged images on frames from **a–d**, **i–l**; **e**1–**h**1, **m**1–**p**1 fluorescence intensity thermal images constructed respectively from **e–h**, **m–p**; **q** mean fluorescence intensity for ROIs in **a, i**: mitochondria; **c, k**: ER; **i**: autophagosome/autolysosome (Apg/Als). n = 6 biologically independent experiments per group, Mean ± SD, the statistical differences between the experimental groups were analyzed by double-tailed Student's t test. When p < 0.05, it was considered to have statistical significance. Scale bar: 2 µm, enlarged images scale bar: 0.5 µm. Source data are available as a Source Data file.

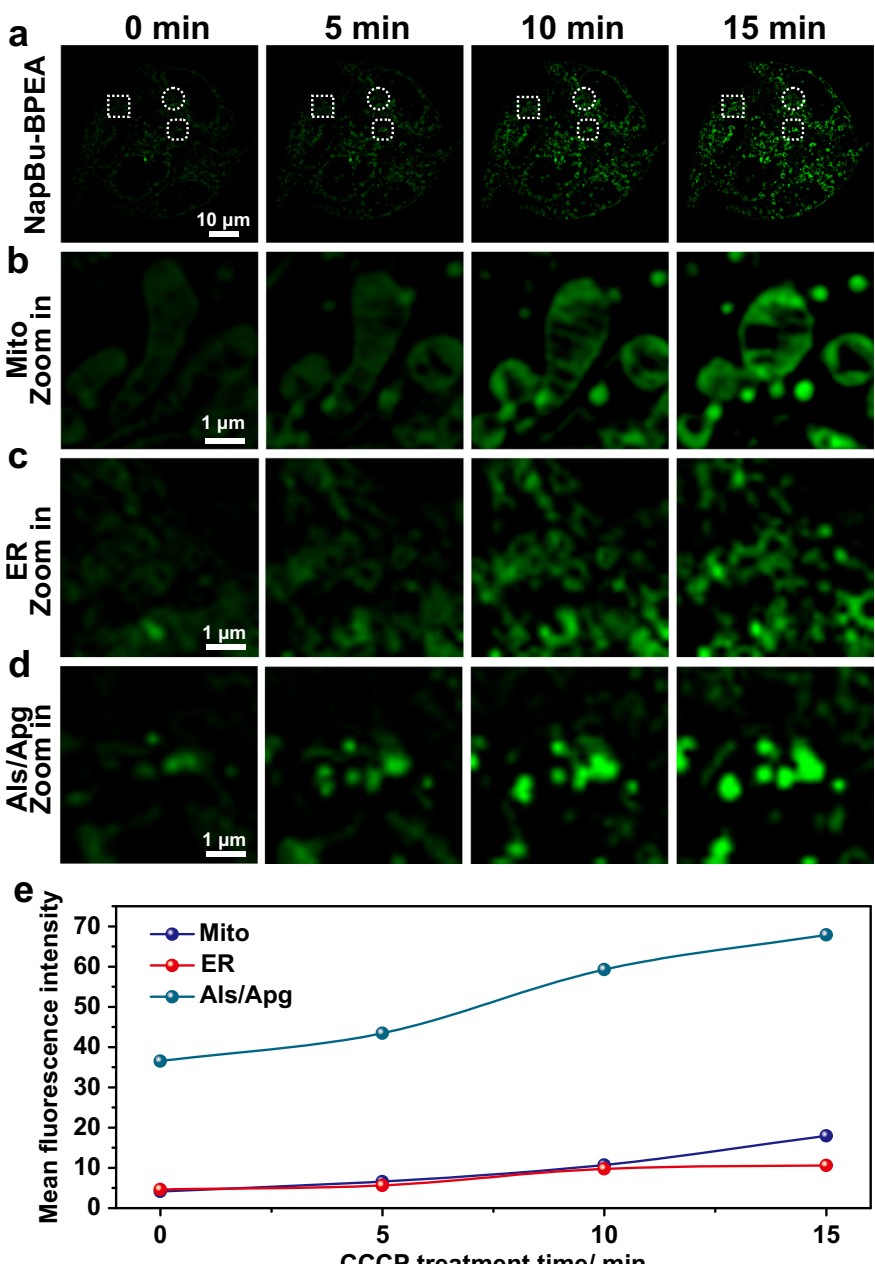

**Fig. 7 NapBu-BPEA enables dynamically track of labile Zn$^{2+}$ in cells undergoing mitophagy via Zn-STIMO. a** Time-lapse SIM images of HeLa cells stained by NapBu-BPEA recorded upon incubation with 20 µM CCCP; and zoom-in images of regions of interest marked with squares (**b**, mitochondria), circles (**c**, ERs), and rounded squares (**d**, autophagosomes/autolysosomes). **e** Temporal profiles of mean fluorescence intensity detected for mitochondria, ERs, and autophagosomes/autolysosomes in HeLa cells showed in **a–d**. Scale bar: 10 µm, enlarged images scale bar: 1 µm. Source data are available as a Source Data file.

more distinct enhancement of labile $Zn^{2+}$ than the mitochondria and ERs (Fig. 7e). This dynamic tracking of labile $Zn^{2+}$ exactly showed the capability of Zn-STIMO via SIM imaging. Thus, we not only proved the concept of Zn-STIMO via morphology-correlated organelle identification, but also employed Zn-STIMO to detect the labile $Zn^{2+}$ fluctuation in various organelles.

**NapBu-BPEA can identify organelles in liver organoids**. Organoids are self-assembled structures formed by differentiating stem cells in vitro[55,56]. Because organoids mimic the structure and function of real organs, they provide simulated human organ models for scientific research and drug development[57,58]. Exploring morphology-correlated organelle identification in organoids via 3D super-resolution imaging may pave the way for tracking a specific organelle and its biomolecules. We used NapBu-BPEA to image organoids via Z-stack mode, and achieved tomographic organelle identification in liver organoids (Fig. 8). Organoids treated with CCCP for 24 h were incubated further with NapBu-BPEA (2 h) and subjected to SIM imaging at different imaging depths. SIM images acquired at imaging depths of 2.4, 6.4, and 14.4 μm all displayed non-uniformly distributed fluorescence in the cytoplasm. Tracking fluorescence along the white lines in Fig. 7e, f showed the imaging resolution to be <200 nm (Fig. 8h, i). This provides SIM imaging the spatial resolution to identify specific organelles in organoids with NapBu–BPEA as a probe. In fact, the ER can be easily identified in the section image according to the visualized morphology (Fig. 8g), and the related fluorescence intensity thermal map revealed the labile $Zn^{2+}$ distribution (Fig. 8j). Therefore, NapBu-BPEA could also be used to track specific organelles, biospecies, and events of interest in human organoids.

## Discussion

Here, we have developed Zn-STIMO through morphology-correlated organelle identification to monitor labile $Zn^{2+}$ fluxes in multiple organelles. For the proof of principle, we rationally designed a probe with a specific, reversible, and turn-on response to $Zn^{2+}$ and with multi-organelle distribution. The success of NapBu-BPEA implies that regulating the lipophilicity of fluorophores without a specific organelle-targeting group might be an effective strategy to alter the probe's subcellular accumulation behavior.

Using the intracellular probe, we observed labile $Zn^{2+}$ enhancement in both the mitochondria and ER during CCCP-inducing mitophagy in HeLa cells. However, the extent of labile $Zn^{2+}$ enhancement was different in the mitochondria and ER, and the autophagosomes and autolysosomes showed the highest labile $Zn^{2+}$ release. The high level of labile $Zn^{2+}$ in the autophagosomes and autolysosomes might be caused by the degradation of $Zn^{2+}$-containing proteins, similar to tamoxifen-induced autophagy releasing labile $Zn^{2+}$ into autophagosomes and lysosomes in MCF-7 cells[59].

By contrast, we observed a decrease in the labile $Zn^{2+}$ level for HeLa cells undergoing starvation-induced autophagy. The different labile $Zn^{2+}$ response between CCCP- and starvation-autophagy suggests that autophagy might undergo pathways associated with different $Zn^{2+}$ buffering, transporting, and sensing proteins to regulate $Zn^{2+}$ homeostasis. Mitochondria, as the main redox site in cells, are rich in reactive oxygen species (ROS). CCCP treatment will not only damage the mitochondrial membrane, but also increase the ROS level, resulting in ROS entering the cytoplasm and stimulating the release of $Zn^{2+}$ from metallothionein[46,60–62]. For starvation-induced autophagy without mitochondrial membrane damage, it has also been reported that nutrient starvation scavenges ROS through activating superoxide

dismutase 2 (SOD2). Therefore, metallothionein can bind more $Zn^{2+}$, resulting in a decrease in intracellular labile $Zn^{2+}$ level[63]. Zn-STIMO in cells undergoing autophagy also revealed a non-uniform distribution of labile $Zn^{2+}$ in both ER and mitochondrion. Therefore, Zn-STIMO via SIM imaging could be used to study the effect of compartmentalized $Zn^{2+}$ distribution on organelle function during autophagy. Furthermore, the probe can facilitate SIM imaging in organoids, which stands to expand current understanding of $Zn^{2+}$-related biology in more complex biological models.

Zn-STIMO via SIM imaging does not involve sophisticated instruments or complicated algorithms, and is compatible with normal fluorophores. Compared to the previously reported method of simultaneously monitoring $Zn^{2+}$ with a series of targeted FRET sensors[21], Zn-STIMO through morphology-correlated organelle identification can distinguish $Zn^{2+}$ fluctuations in multiple organelles with only one probe, which can significantly reduce labeling damage and minimize fluorescent crosstalk. Together, the current results demonstrated that this super-resolution morphology-correlated organelle identification is an effective strategy for simultaneous tracking in multiple organelles in living cells, and can also be applied to track other biospecies and events in multiple subcellular organelles. With all these, we expect this method would transform the study of subcellular $Zn^{2+}$ biology.

Meanwhile, the morphology-based organelle discrimination is mainly for organelles with obvious morphological features such as mitochondria and endosome reticula. For the vesicle-like lysosomes and autolysosomes/autophagosomes, the discrimination requires additional tool for verification. In our study, we have initially carried out the co-localization experiments using commercial dyes, which verified round fluorescence spots in cells under normal condition as lysosomes (Fig. 3) and similar fluorescence spots in cells under autophagy condition as autolysosomes/autophagosomes (Fig. 5). To study organelles with similar features, staining with labeled organelle-specific protein for further confirmation would increase the reliability of Zn-STIMO. Since we only use one fluorescent channel, there would be enough remaining channels for the simultaneous imaging of labeled proteins. Furthermore, together with artificial intelligence, more accurate identification of various organelles will be achieved.

Although the Zn-STIMO based on super-resolution morphology-correlated organelle identification can detect biospecies changes in subcellular organelles, it remains difficult to apply the method to smaller organelles (e.g., synaptic vesicles) due to the limitation of SIM resolution. Therefore, if the technique is combined with higher resolution imaging techniques—stimulated emission depletion (STED) microscopy, for example—more detailed subcellular structures and subtle changes at the species level could be observable. Moreover, the current Zn-STIMO is based on a $Zn^{2+}$ probe showing turn-on response, and interference from the dynamic distribution of probe can be expected. Therefore, developing ratiometric $Zn^{2+}$ probe for Zn-STIMO is still undergoing in our lab to overcome this disadvantage.

## Methods

**Materials and instrument**. All solvents and reagents are of analytical grade and used without further purification. 4-Bromo-1, 8-naphthalic anhydride, *n*-butylamine, ethylenediamine, picolyl chloride, and $Ru(bpy)_3^{2+}$ were purchased from Energy Chemical Inc (Shanghai, China). $KCl$, $CaCl_2$, $MgCl_2$, $NaCl$, $FeSO_4$, $FeCl_3$, $Zn(NO_3)_2$, $NiCl_2$, $CdCl_2$, $MnCl_2$, $BaCl_2$, $CrCl_3$, $Al_3(SO_4)_2$, and $Pb(NO_3)_2$ were purchased from Sinopharm Chemical Reagent (Nanjing, China). Carbonyl cyanide m-chlorophenylhydrazone (CCCP), ethylenebis (oxyethylenenitrilo) tetraacetic acid (EGTA) and 2-[4-(2-hydroxyethyl)-1-piperazinyl]ethanesulfonic acid (HEPES) were purchased from Sigma (Shanghai, China). Pyrithione sodium salt, N, N, N′, N′-tetrakis(2-pyridylmethyl)ethanediamine were obtained from Fisher Scientific Inc (OH, USA). MitoTracker™ Green FM (MTG), MitoTracker™ DeepRed FM, LysoTracker™ Red DND-99 (LTR), ER-Tracker™ Red and Hoechst 33258 were

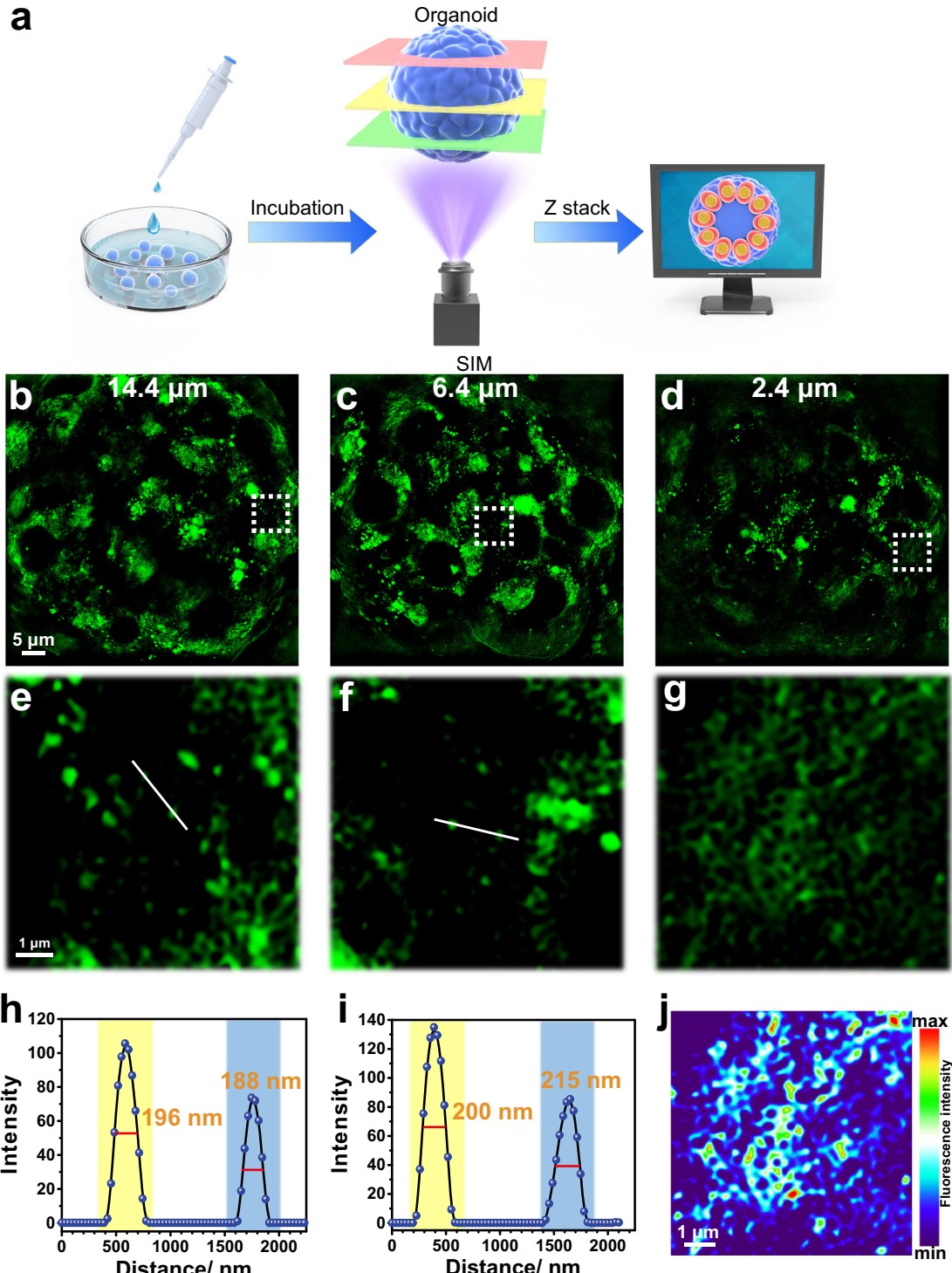

**Fig. 8 NapBu-BPEA can identify organelles in liver organoids.** SIM imaging of organoids treated with CCCP for 24 h. **a** Schematic diagram of Z-stack SIM imaging for organoid; **b–d** SIM images of organoids recorded at different imaging depths; **e–g** enlarged images for the regions of interest showed in the frames in **b–d**; **h, i** fluorescence intensity profiles along the white lines in **e, f**; **j** fluorescence intensity thermal image of **g** constructed with ImageJ. Scale bar: 5 µm, enlarged images scale bar: 1 µm.

purchased from Invitrogen (OH, USA). Autophagosome Detection dye (DAPRed) and Cytoxicity LDH Assy Kit-WST were purchased from Dojindo (Washington, USA). The cell culture medium, Dulbecco's Modified Eagle Medium (DMEM) and Earle's Balanced Salt Solution (EBSS, calcium, magnesium, phenol red) were bought from Gibco (OH, USA).

The $^1$H NMR and $^{13}$C NMR spectra were determined with a 400 M Bruker spectrometer with TMS as internal standard. High-Resolution Mass spectrometric data were recorded on an Agilent 6540 Q-TOF mass spectrometer. The UV-Vis and fluorescence spectra were performed on PerkinElmer Lambda 35 spectrophotometer

and Horiba FM-4 fluorophotometer. The cell imaging was carried out by Nikon N-SIM system.

**Synthesis of 4-bromo-$N$-n-butyl-1, 8-naphthalimide (1)**[64]. $^1$H NMR (400 MHz, Chloroform-$d$) δ 8.66 (dd, $J$ = 7.3, 1.1 Hz, 1H), 8.57 (dd, $J$ = 8.5, 1.2 Hz, 1H), 8.42 (d, $J$ = 7.8 Hz, 1H), 8.04 (d, $J$ = 7.9 Hz, 1H), 7.85 (dd, $J$ = 8.5, 7.3 Hz, 1H), 4.24–4.06 (m, 2H), 1.83–1.62 (m, 2H), 1.52–1.34 (m, 2H), 0.98 (t, $J$ = 7.3 Hz, 3H).

## Synthesis of *N*-n-butyl-4-(aminoethylene) amino-1, 8-naphthalimide (2)[64].

$^1$H NMR (400 MHz, Chloroform-*d*) δ 8.59 (dd, *J* = 7.3, 1.1 Hz, 1H), 8.47 (d, *J* = 8.4 Hz, 1H), 8.17 (dd, *J* = 8.5, 1.1 Hz, 1H), 7.63 (dd, *J* = 8.4, 7.3 Hz, 1H), 6.71 (d, *J* = 8.5 Hz, 1H), 6.14 (s, 1H), 4.21–4.12 (m, 2H), 3.42 (q, *J* = 5.3 Hz, 2H), 3.18 (dd, *J* = 6.7, 4.9 Hz, 2H), 1.79–1.69 (m, 2H), 1.51–1.38 (m, 2H), 0.97 (t, *J* = 7.3 Hz, 3H).

## Synthesis of 2,4-(bis(pyridin-2-ylmethyl)aminoethyl)amino-*N*-n-butyl-1,8-naphthalimide (NapBu–BPEA)[65].

In all, 200 mg (0.64 mmol) compound 2, 300 mg K$_2$CO$_3$, and 330 mg (2.6 mmol) picolyl chloride were dissolved in 20 mL dry ethanol. The mixture was stirred and refluxed for 10 h under N$_2$. The reaction process was monitored by TLC. The solvent was evaporated by under reduced pressure, once the reaction was completed. Then the crude product was purified by silica gel column chromatography (CH$_2$Cl$_2$: CH$_3$OH = 20:1) to obtain a yellow solid with 24.6% yield (70 mg). $^1$H NMR (400 MHz, Methanol-*d*$_4$) δ 8.59 (dd, *J* = 8.4, 1.2 Hz, 1H), 8.50 (d, *J* = 7.3 Hz, 1H), 8.42 (dd, *J* = 5.0, 1.3 Hz, 2H), 8.17 (d, *J* = 8.5 Hz, 1H), 7.68 (dd, *J* = 8.4, 7.3 Hz, 1H), 7.51–7.47 (m, 4H), 7.14 (ddd, *J* = 6.6, 5.2, 2.4 Hz, 2H), 6.51 (d, *J* = 8.5 Hz, 1H), 4.15–4.03 (m, 2H), 3.88 (s, 4H), 3.52 (t, *J* = 5.9 Hz, 2H), 2.92 (t, *J* = 5.9 Hz, 2H), 1.67 (tt, *J* = 8.0, 6.4 Hz, 2H), 1.42 (q, *J* = 7.5 Hz, 2H), and 0.99 (t, *J* = 7.3 Hz, 3H). $^{13}$C NMR (101 MHz, Methanol-*d*$_4$) δ 166.25, 165.72, 160.27, 152.30, 149.61, 138.51, 135.86, 132.22, 131.27, 129.50, 125.49, 125.25, 123.79, 123.46, 122.00, 109.23, 105.12, 61.16, 53.19, 49.68, 49.54, 49.47, 49.32, 49.25, 49.04, 48.83, 48.69, 48.62, 48.40, 41.88, 40.84, 31.43, 21.42, and 14.28. HR-MS(positive mode): Calcd. 494.2551, Found. 494.2649.

## Spectroscopic study.

The stock solution of NapBu–BPEA (10 mM) was prepared with DMSO of HPLC pure grade, and stored at −20 °C. The work solutions of this probe for spectroscopic study were prepared by diluting this stock solution to the final concentration of 10 μM with 3 mL HEPES buffer (50 mM, 100 mM KNO$_3$, 10% DMSO, pH 7.2) in quartz cuvettes with 1 cm path lengths. All fluorescent spectra were recorded upon excitation at 450 nm.

The Zn$^{2+}$ titration UV-vis and fluorescent spectra were recorded by adding aliquots of Zn$^{2+}$ solution into the probe work solution (10 μM). The detection limit of NapBu–BPEA was determined by recording the fluorescence spectra of NapBu–BPEA for six times to obtain the background noise (σ). The probe's fluorescent sensing selectivity of NapBu–BPEA were determined by recording fluorescence spectra after adding metal cation (1000 eq K$^+$, Na$^+$, Ca$^{2+}$,Mg$^{2+}$; 1 eq Cd$^{2+}$, Ni$^{2+}$, Cr$^{3+}$, Pb$^{2+}$, Al$^{3+}$, Co$^{2+}$, Fe$^{2+}$, Fe$^{3+}$, Ba$^{2+}$, Mn$^{2+}$) to the NapBu–BPEA solution. The fluorescence quantum yields were determined by using Ru(bpy)$_3$$^{2+}$ (Φ = 0.04) in DMSO/HEPES (*v:v*, 1:9) as the reference. To determine binding constant, various amounts of Zn(NO$_3$)$_2$ (0~9 mM) were added to NapBu–BPEA solution buffered with DMSO/HEPES (50 mM, pH 7.21, 100 mM KNO$_3$) containing 10 mM EGTA. The probe's fluorescence spectra at different pH were measured by recording the spectra of NapBu–BPEA solution in the presence of Zn$^{2+}$ at different pH adjusted with KOH and HCl.

## Cell culture.

Wild-type, FIP 200 and ATG KO HeLa cell lines were gifted from Dr. Jun-Lin Guan's lab (University of Cincinnati). Cells were cultured in DMEM supplemented with 10% FBS and 100 U/mL penicillin–streptomycin (Gibco) in 5% CO$_2$ incubator at 37 °C.

## Organoid culture.

Human induced pluripotent stem cells (iPSCs) were differentiated into foregut using previously described method[56]. In brief, hiPSCs were detached by Accutase (Thermo Fisher Scientific Inc., MA, USA) and were seeded on Laminin coated tissue culture plate with 100,000 cells/cm$^2$. Medium was changed to RPMI 1640 medium (Life Technologies) containing 100 ng/mL Activin A (R&D Systems) and 50 ng/mL bone morphogenetic protein 4 (BMP4; R&D Systems) at day 1, 100 ng/mL Activin A and 0.2% fetal calf serum (FCS; Thermo Fisher Scientific Inc.) at day 2, and 100 ng/mL Activin A and 2% FCS at day 3. On day 4–6, cells were cultured in Advanced DMEM/F12 (Thermo Fisher Scientific Inc.) with B27 (Life Technologies) and N2 (Gibco, CA, USA) containing 500 ng/mL fibroblast growth factor (FGF4; R&D Systems) and 3 μM CHIR99021 (Stemgent, MA, USA). Cells were maintained at 37 °C in 5% CO$_2$ with 95% air and the medium was replaced every day. The foregut cells were detached by Accutase and then centrifuged at 1200 rpm for 3 min. Cells were resuspended in Matrigel (Corning, In., NY, USA). A total of 100,000 cells were embedded in 50 μL Matrigel drop on the dishes in organoid formation media with 5 factors for 4 days. After organoid formation, the media was switched to liver specification media for 4 days. After the liver specification step, organoids were harvested from Matrigel by scratching and pipetting. Then organoids were re-embedded in Matrigel on the Ultra-low attached plate (Corning) in liver maturation media for 10 days. Cultures for HLO induction were maintained at 37 °C in 5% CO$_2$ with 95% air and the medium was added every 2 days.

## Nikon SIM super-resolution imaging.

The SIM images were acquired using a Nikon N-SIM system. The blue imaging channel for Hoechst 33258 with emission bandwidth at 420–495 nm upon excitation at 405 nm, the green imaging channel for NapBu-BPEA with emission bandwidth at 500–550 nm upon excitation at

488 nm, the red imaging channel for ER-Tracker Red, LysoTracker Red and DAPRed with emission bandwidth at 570–640 nm upon excitation at 561 nm, the magenta imaging channel for MitoTracker Deep Red with emission bandwidth at 660–735 nm upon excitation at 640 nm were utilized. The imaging data analysis and thermal map construction were performed via analysis with ImageJ.

The co-localization experiments were performed with a dual-channel mode. HeLa cells were stained by NapBu–BPEA (10 μM, 1 h) and then incubated with Mito-marker Deep Red (0.5 μM, 30 min), Lysotracker Red (0.1 μM, 30 min), ER-Tracker Red (1 μM, 30 min), and Hoechst 33258 (1 μg/mL, 30 min), respectively. The Pearson's correlation coefficient was calculated using Cellprofiler with co-localization module.

The intracellular Zn$^{2+}$ level in autophagy was imaged in HeLa cells. Prior to CCCP (10 μM, 24 h) or EBSS treatment (10 μM, 24 h), the cells were stained with DAPRed (1 μM, 30 min). The cells were finally stained by NapBu–BPEA (10 μM, 1 h) before SIM imaging.

The fresh organoids were transferred into petri dish. After 10 μM CCCP treatment for 24 h, the organoids were incubated with 10 μM NapBu–BPEA for 2 h. Then the organoids were imaged with z stack at different depths.

## Cell viability determination via WST assay.

The suspension of HeLa cells diluted with 50 μL DMEM was planted into 96-well plate. The inoculated cells were pre-cultured overnight in 96-well plate and replaced with a new 50 μL DMEM. 50 μL DMEM containing different concentrations of NapBu–BPEA was added and cultured in CO$_2$ incubator at 37 °C for 24 h. After 10 μL Lysis Buffer was added to the high contrast wells, 30 min was cultured in the CO$_2$ incubator at 37 °C. After 100 μL Working Solution was added to each well, it was cultured for 0.5 h under dark and room temperature. After 50 μL Stop Solution was added to each well, the absorbance of 490 nm was determined immediately by a microplate reader (Thermomax, Molecular Devices).

## Data analysis.

All data were analyzed and statistically calculated using Microsoft Excel 2016 software (Microsoft, Redmond, WA). The results are expressed as mean ± standard deviation (SD) unless otherwise stated. The statistical differences between the experimental groups were analyzed by double-tailed Student's *t*-test. When $p < 0.05$, it was considered to have statistical significance. All statistical graphs were performed using Origin 2016 (OriginLab Corporation, MA, USA).

## Statistics and reproducibility.

Each experiment was repeated at least three times independently with similar results. All images shown are representative results from biological replicates.

## Reporting summary.

Further information on research design is available in the Nature Research Reporting Summary linked to this article.

## Data availability

All data supporting the findings of this study are available ether in the article and/or its Supplementary Information files or from the authors upon reasonable request. Source data are provided with this paper.

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

## Acknowledgements

Y.C., Z.G., and W.H. were supported by the Natural Science Foundation of China (Grants 21977044, 21731004, 21907050 and 91953201), the Natural Science Foundation of Jiangsu Province (BK20190282) and the Excellent Research Program of Nanjing University (ZYJH004). T.T. is a New York Stem Cell Foundation – Robertson Investigator. J.D. was supported by the Department of Cancer Biology, University of Cincinnati College of Medicine. We thank Dr. Yi Cao at Nanjing University for fruitful discussions and Dr. Katrina Woolcock for editing services.

## Author contributions

H.F. collected all 3D-SIM super-resolution microscopy data. H.F. and Q.C. analyzed and processed the SIM data. M.H. and Z.T. cultured the cells. S.G. and H.F. synthesized the probe and characterized the spectral property of the probe. C.L. cultured the liver organoids, T.T. conceived the organoids experiment. J.G. designed the autophagy-related protocol. M.L. and C.W. calculated the molecule structures. Y.C., Z.G., W.H., and J.D. conceived the project, designed the experiments, and wrote the manuscript with the help of all authors.

## Competing interests

The authors declare no competing interests.
