## [Peer Review File · Nature Communications]

REVIEWER COMMENTS

Reviewer #1 (Remarks to the Author):

This work developed an organelle-nonspecific Zn²⁺ fluorescent probe by regulating the lipophilicity of existing Zn²⁺ probes, and employed SIM super-resolution imaging to distinguish different organelles upon their corresponding characteristic morphology. This idea is able to potentially overcome some shortcomings of current Zn²⁺ probes. However, there are some concerns about the performance of the NapBu-BPEA probe, and the feasibility of this idea in practical experiments.

(a) It is known that the majority of Zn ion are binding with numerous proteins functioning in cytosol and various organelles (Outten CE, O'Halloran TV (2001) *Science* 292:2488–2492; Vinkenberg JL, Nicolson TJ, Bellomo EA et al (2009) *Nat Methods* 6: 737–740; Qin Y, Miranda JG, Stoddard CI et al (2013) *ACS Chem Biol* 8:2366–2371), so that the concentration of free Zn is quite low, ranging from pico- to nano-molar, for instance, 0.14 pM in mitochondria (Besnard P, Niot I, Poirier H et al (2002)), 0.2 pM in mitochondrial matrix (McCranor BJ, Bozym RA, Vitolo MI et al (2012) *J Bioenerg Biomembr* 44:253–263), 0.9 pM–5 nM in ER and 0.2 pM in Golgi (Qin Y, Dittmer PJ, Park JG et al (2011); *Proc Natl Acad Sci USA* 108: 7351–7356). However, the authors demonstrated that the dissociation constant (K_d) of the Zn²⁺/NapBu-BPEA complex was 4.98 nM (Figures 2d and S7d). Given that K_d of Zn²⁺/NapBu-BPEA is approximately 1000 times higher than the Zn²⁺ concentration in distinct organelles, it raises the concern that whether the NapBu-BPEA is really sensitive enough to detect these ultra-low labile Zn concentrations in living cells.

(b) Although most organelles have their own characteristic morphologies, the vesicle-like shapes are shared by many different organelles, and they undergo constant deforming. For instance, the Golgi-derived vesicles are highly similar with lysosomes, or autophagosomes. It would be hard to discriminate them even using higher resolution methods.

(c) The authors declared that NapBu-BPEA rarely go into nucleus, indicating that NapBu-BPEA's permeability varies among different organelles. This might be reasonable, because distinct organelles have quite different membrane structures and compositions. Therefore, this proposed method actually could not quantitatively evaluate or compare the concentration of Zn²⁺ in different organelles.

Other minor points:

(i) Page 5, because not all readers are familiar with the chemical terms, please provide the definition for logP and logP_{Zn} values.

(ii) In Figure 2a, the concentration of Zn²⁺ is on the y-axis, what is the lowest concentration of Zn²⁺ for the first curve? By my eye, the lowest concentration of Zn²⁺ seems to be 0 μM. If that is true, it means the NapBu-BPEA probe has a relatively high background signal even without Zn²⁺ binding.

Reviewer #2 (Remarks to the Author):

This is an interesting article that proposes the utilization of the novel fluorescent probe NapBu-BPEA to analyze Zn²⁺ in multiple organelles and cells. NapBu-BPEA binds Zn²⁺ in a reversible way with high specificity and exhibits low toxicity and low photobleaching. The conclusions are well supported by the data. The authors provided good quality data. The work can be reproduced but some details need to be provided (see below).

Statistical analysis utilized "was not described" in the manuscript.

There are some issues/questions that could be addressed to improve the quality of the manuscript:

- 1) The K_d of the Zn^{2+} /NapBu-BPEA complex was 4.98 nM. How this K_d value compares to the K_d values of other Zn^{2+} fluorophores and the K_d values of zinc binding proteins? Will NapBu-BPEA take zinc from Metallothioneins or zinc finger proteins?
- 2) In Figure 4a, the fluorescence increases with time upon incubation with 50 μ M ZnPT. The 0 min time point shows no fluorescence in figure 4a. However, in figure 3 the cells show green fluorescence. Was ZnPT used as well for figure 3? What is the difference that causes this result in Figure 4a? Different gain-exposure?
- 3) In Figure 6, it will help to show the co-stain MitoTracker and DAPI. It will be helpful to include a publication that confirms that DAPI labels autophagosomes and autolysosomes in a specific way.
- 4) Under Discussion: "The different labile Zn^{2+} response between CCCP- and starvation-autophagy suggests that autophagy might undergo pathways associated with different Zn^{2+} buffering, transporting and sensing proteins to regulate Zn^{2+} homeostasis." This could be explained better.

Reviewer #3 (Remarks to the Author):

He, Diao, Chen and co-authors reported a new method to simultaneously track Zn^{2+} in living cells based on the super-resolution morphology correlated organelles. It looks very interesting, especially with very fantastic colorful imaging with high-resolution in living cells. However, in my opinion, this is a kind of "old wine" in a new concept created by themselves. First, tracking dynamic Zn^{2+} ion in living cells is important in understanding Zn homeostasis in bioinorganic chemistry, so development of Zn sensor is consequently important. There are many Zn fluorescence sensors reported in recent years and most of them are based on ICT mechanism. The so-called simultaneous Zn^{2+} tracking in multiple organelles is a kind of Zn sensor essentially with non-specific organelle targeting. In this aspect, the design of NapBu-BPEA is not novel. Second, the proof-of-concept experiments have been done mostly using CCCP inducing Zn^{2+} release in mitochondria. However, CCCP treatment is very complicated, as the authors also indicated. At current stage, it is lack of evidence to demonstrate the green fluorescence species is Zn NapBu-BPEA in living cells, where I suggest to perform X-ray fluorescence experiments. This is important for detecting intracellular Zn^{2+} . Many groups reported good Zn sensors in live cells have been eventually proven to be "turn on" by other species than Zn^{2+} . This should be careful.

Third, tracking dynamic Zn^{2+} in living cells is very attractive. However, I did not read this interesting part about monitoring dynamic process. If there is no such study, I think this is a paper about sensor design to catch Zn^{2+} using an advanced fluorescence microscopy.

Suggestions:

1. Pls. give a general scheme or figure to demonstrate the concept of Zn-STIMO, which is important to clarify.
2. Pls. reorganize the manuscript and reduce the tedious part such as the design of probe (page 7).
3. Pls. delete interpretation or explanation part you do not have strong evidences to support.

Responses to Reviewers' Comments:

Our responses are highlighted in blue. The revision of the text is highlighted in yellow color.

Reviewer #1:

Comment and Question 1 (C&Q 1): This work developed an organelle-nonspecific Zn^{2+} fluorescent probe by regulating the lipophilicity of existing Zn^{2+} probes, and employed SIM super-resolution imaging to distinguish different organelles upon their corresponding characteristic morphology. This idea is able to potentially overcome some shortcomings of current Zn^{2+} probes. However, there are some concerns about the performance of the **NapBu-BPEA** probe, and the feasibility of this idea in practical experiments.

Response and Answer 1 (R&A 1): Thanks a lot for your helpful comments and suggestions. We have revised the manuscript carefully according to your concerns and other referees' comments.

C&Q 2: It is known that the majority of Zn ion are binding with numerous proteins functioning in cytosol and various organelles (Outten CE, O'Halloran TV (2001) *Science* 292:2488–2492; Vinkenborg JL, Nicolson TJ, Bellomo EA et al (2009) *Nat Methods* 6:737–740; Qin Y, Miranda JG, Stoddard CI et al (2013) *ACS Chem Biol* 8:2366–2371), so that the concentration of free Zn is quite low, ranging from pico- to nano-molar, for instance, 0.14 pM in mitochondria (Besnard P, Niot I, Poirier H et al (2002)), 0.2 pM in mitochondrial matrix (McCranor BJ, Bozym RA, Vitolo MI et al (2012) *J Bioenerg Biomembr* 44:253–263), 0.9 pM–5 nM in ER and 0.2 pM in Golgi (Qin Y, Dittmer PJ, Park JG et al (2011); *Proc Natl Acad Sci USA* 108:7351–7356). However, the authors demonstrated that the dissociation constant (K_d) of the Zn^{2+} /NapBu-BPEA complex was 4.98 nM (Figures 2d and S7d). Given that K_d of Zn^{2+} /NapBu-BPEA is approximately 1000 times higher than the Zn^{2+} concentration in distinct organelles, it raises the concern that whether the NapBu-BPEA is really sensitive enough to detect these ultra-low labile Zn concentrations in living cells.

R&A 2: Thanks for the valuable comment. It is true that most Zn^{2+} would bind with proteins, and labile Zn^{2+} is normally very low (the picomolar to low nanomolar). However, various stimuli will lead to extra- and intracellular Zn^{2+} release to fulfill the signaling function, and the labile Zn^{2+} level may attain to micromolar. (J. J. Hwang et al., *J. Neurosci.* **2008**, 28, 3114; S. L. Sensi et al., *PNAS* **2003**, 100, 6157; S. A. Abiria et al., *PNAS* **2017**, 114, E6079; W. Lin et al., *JACS* **2013**, 135, 13512; T. Miki et al., *Nat. Methods* **2016**, 13, 931). Meanwhile, there are labile Zn^{2+} pools in some specific cells, such as neurons,

oocytes, etc., wherein the Zn^{2+} concentration can even reach the μM level (L. A. Finney et al., *Science* **2003**, 300, 931). Moreover, the compartmentalized distribution of intracellular labile Zn^{2+} has been confirmed to show variable labile Zn^{2+} level (Itsumura et al., *Physiol. Rev.* **2015**, 95, 749). It is clear that labile Zn^{2+} levels is variable and displays as specific spatiotemporal distribution patterns, and labile Zn^{2+} possesses the μM level in Zn^{2+} pool or upon stimulation under various stresses, besides the well-known pico- and nanomolar level. All these variable Zn^{2+} levels demands probes of different Zn^{2+} affinity to sense these Zn^{2+} level effectively.

A brief survey on the organelle-targeting Zn^{2+} probes showed that most of these probes possess the K_d values ranging from 0.2 to 70 nM (new Table S3). Our probe **NapBu-BPEA** showed a Zn^{2+} K_d value locating exactly in this range, indicating its Zn^{2+} affinity is suitable for sensing labile Zn^{2+} in organelles just as these reported probes. Furthermore, Figures 5 and new Figure S15 showed also that our probe is sensitive enough to detect the reversible change of Zn^{2+} level during autophagy. (See page 9 in the revised version)

It should be noted that the labile Zn^{2+} level in organelles is still an open question (T. Kambe et al., *Physiol. Rev.* **2015**, 95,749), and developing a series of probes showing variable Zn^{2+} affinity is of great significance to clarify labile Zn^{2+} level in different organelles. We are still developing probes of different Zn^{2+} affinities including those of high Zn^{2+} binding constants to realize more accurate imaging for variable labile Zn^{2+} level in multi-organelles.

Table S3. Organelle-targeting Zn^{2+} probes and their Zn^{2+} affinities.

Probe	K_d / nM	Organelles	Cells	References
RhodZin-3 AM	~65	Mitochondria	Neurons	S. L. Sensi et al., P. Natl. Acad. Sci. USA 2003 , 100, 6157.
ZP1-TPP	0.6	Mitochondria	HeLa cells	W. Chyan et al., P. Natl. Acad. Sci. USA 2014 , 111, 143.
DQZn4	0.2	Lysosome	NIH 3T3 cells	L. Xue et al., Inorg. Chem. 2012 , 51, 10842.
LysoDPP-C4	1.91	Lysosome	HeLa cells	C. Du et al., Chem. Sci. 2019 , 10, 5699.
Probe 9	3.5	ER	HeLa cells	L. Fang et al., Chem. Sci. 2019 , 10, 10881.
SZnC	1.7	Golgi	HeLa cells	H. Singh et al., Chem. Commun. 2015 , 51, 12099.

C&Q 3: Although most organelles have their own characteristic morphologies, the vesicle-like shapes are shared by many different organelles, and they undergo constant deforming. For instance, the Golgi-derived vesicles are highly similar with lysosomes, or autophagosomes. It would be hard to discriminate them even using higher resolution methods.

R&A 3: We agree with the referee's comment. For Zn-STIMO proposed in our manuscript, the morphology-based organelle discrimination is mainly for organelles with obvious morphological features such as mitochondria and endosome reticula. For the vesicle-like lysosomes and autolysosomes/autophagosomes, the discrimination requires additional tool for verification. In our study, we have initially carried out the co-localization experiments using commercial dyes, which verified that dot fluorescence in cells under normal condition is from lysosomes (Figure 3) and dot fluorescence in cells under autophagy condition is from autolysosomes/autophagosomes (Figure 5).

To study other organelles with similar features, staining with an organelle-specific labeling proteins for further confirmation might be a reliable tool. Since our probe only occupies one fluorescent channel, there are enough remaining channels for the simultaneous imaging of labeling proteins. In this way, Zn-STIMO can still be realized easily.

C&Q 4: The authors declared that NapBu-BPEA rarely go into nucleus, indicating that NapBu-BPEA's permeability varies among different organelles. This might be reasonable, because distinct organelles have quite different membrane structures and compositions. Therefore, this proposed method actually could not quantitatively evaluate or compare the concentration of Zn^{2+} in different organelles.

R&A 4: We appreciate your meaningful comment. Indeed, the cumulative concentrations of the probe in different organelles could be different, and we also fully aware this issue. Therefore, Zn-STIMO with NapBu-BPEA is able to visualize labile Zn^{2+} fluctuations in different organelles of cells in normal or stimulated conditions, but could not quantitatively evaluate or compare the concentration of Zn^{2+} in different organelles. To realize Zn-STIMO for labile Zn^{2+} quantification and concentration comparison in multiple organelles, we are now developing ratiometric Zn^{2+} probes accumulating in multiple organelles for Zn-STIMO. Hope we could make advance in the near future. Another alternative is lifetime probes for Zn^{2+} , in this case, the SIM instrument enabling rapid fluorescence lifetime discrimination is required. We will pay attention to the development of new SIM instrument.

Other minor points:

C&Q 5: Page 5, because not all readers are familiar with the chemical terms, please provide the definition for logP and logPZn values.

R&A 5: Thanks for the reminding. We have given the definitions in the revised manuscript and highlighted in yellow.

“The calculation of $\log P$ (partition coefficient of free probe in n-octanol/water) and $\log P_{Zn}$ (partition coefficient of probe’s zinc complex in n-octanol/water) for probes **Naph-BPEA**, **NapEt-BPEA** and **NapBu-BPEA** disclosed that the **NapBu-BPEA** possessed the highest $\log P$ and $\log P_{Zn}$ values among the three, we hypothesized that this probe has the desired subcellular distribution behavior, although **NapEt-BPEA** shows the similar accumulation behavior to that of **Naph-BPEA**.”

C&Q 6: In Figure 2a, the concentration of Zn^{2+} is on the y-axis, what is the lowest concentration of Zn^{2+} for the first curve? By my eye, the lowest concentration of Zn^{2+} seems to be 0 μM . If that is true, it means the NapBu-BPEA probe has a relatively high background signal even without Zn^{2+} binding.

R&A 6: Thanks for the comment. It is real that there is background fluorescence in the imaging with this probe, since there is only moderate PET effect from the probe’s chelator to fluorophore to quench the fluorescence of apo probe. However, Zn-STIMO is a method based on the discrimination of organelle morphologies. The probe’s background fluorescence favors the visualization of initial morphologies of various organelles, and help to realize the tracking of labile Zn^{2+} fluctuation in multiple organelles.

Reviewer #2:

C&Q 7: This is an interesting article that proposes the utilization of the novel fluorescent probe NapBu-BPEA to analyze Zn^{2+} in multiple organelles and cells. NapBu-BPEA binds Zn^{2+} in a reversible way with high specificity and exhibits low toxicity and low photobleaching. The conclusions are well supported by the data. The authors provided good quality data. The work can be reproduced but some details need to be provided (see below).

R&A 7: Thanks for the positive comments. We have revised the manuscript according to your advices and other referee's comments.

C&Q 8: Statistical analysis utilized "was not described" in the manuscript.

R&A 8: Thank you for your reminder and sorry for the carelessness. We have added the description of t-test method in the Methods section of the revised manuscript and highlighted in yellow color.

Statistical analysis. All data were analyzed and statistically calculated using Microsoft Excel 2016 software (Microsoft, Redmond, WA). The results are expressed as mean \pm standard deviation (SD) unless otherwise stated. The statistical differences between the experimental groups were analyzed by double-tailed student t-test. When $p < 0.05$, it was considered to have statistical significance. All statistical graphs were performed using Origin 2016 (OriginLab Corporation, MA, USA).

C&Q 9: There are some issues/questions that could be addressed to improve the quality of the manuscript: The K_d of the Zn^{2+} /NapBu-BPEA complex was 4.98 nM. How this K_d value compares to the K_d values of other Zn^{2+} fluorophores and the K_d values of zinc binding proteins? Will NapBu-BPEA take zinc from Metallothioneins or zinc finger proteins?

R&A 9: Thank you for the constructive comment. The K_d values of most reported Zn^{2+} probes of organelle-targeting ability were listed in Table S3 (See **R&A 2**). Information from Table S3 indicated that the K_d value of probe NapBu-BPEA is comparable to those of many reported Zn^{2+} probes.

As the Zn^{2+} buffer proteins, metallothioneins (MTs), are labile zinc source in cells. They normally bind with Zn^{2+} showing a K_d value of 0.32 pM at pH=7.4. Therefore, probe NapBu-BPEA is difficult to take Zn^{2+} from MTs. However, this probe is able to sense labile Zn^{2+} release from MTs induced by various stresses and stimulations (C. Jacob et al., *P. Natl. Acad. Sci. USA* **1998**, 95, 3489).

C&Q 10: In Figure 4a, the fluorescence increases with time upon incubation with 50 μ M ZnPT. The 0

min time point show no fluorescence in figure 4a. However, in figure 3 the cells show green fluorescence. Was ZnPT used as well for figure 3? What is the difference that causes this result in Figure 4a? Different gain-exposure?

R&A 10: Thanks for the comment. There was no ZnPT treatment for Figure 3. Just as the reviewer has pointed out that the different imaging conditions were the origin for the different fluorescence observed between cells undergoing ZnPT at 0 min (Figure 4a) and normal cells without ZnPT treatment (Figure 3). To disclose the probe's subcellular distribution pattern (Figure 3), we used higher excitation power and longer exposure time to visualize more organelles in normal cells. For experiments corresponding to Figure 4a, we utilized lower excitation power and shorter exposure time to make the initial cells show almost no fluorescence, with the purpose to demonstrate the whole dynamic process of intracellular Zn^{2+} enhancement induced by ZnPT.

C&Q 11: In Figure 6, it will help to show the co-stain MitoTracker and DAPRed. It will be helpful to include a publication that confirms that DAPRed labels autophagosomes and autolysosomes in a specific way.

R&A 11: Thanks for the helpful advice. The technical manual of DAPRed indicates this dye is incorporated into the autophagosome during double-membrane formation via structural features, and then emits fluorescence under hydrophobic conditions. This has been briefly discussed in the revised manuscript. The webpage for this manual together with a reference paper of DAPGreen showing the same imaging mechanism for autophagy have been cited. See Ref. 44, 45 in the revised version.

C&Q 12: Under Discussion: “The different labile Zn^{2+} response between CCCP- and starvation-autophagy suggests that autophagy might undergo pathways associated with different Zn^{2+} buffering, transporting and sensing proteins to regulate Zn^{2+} homeostasis.” This could be explained better.

R&A 12: Thanks for the helpful suggestion. We have modified the discussion as the follow: “Mitochondria, as the main redox site in cells, are rich in reactive oxygen species (ROS). CCCP treatment will not only damage the mitochondrial membrane, but also induce oxidative stress and mitochondrial ROS enhancement, and finally elevate cytosolic ROS level to stimulate labile Zn^{2+} release from Zn^{2+} buffering proteins, metallothioneins (MTs).^{42,56-58} The starvation-induced autophagy does not induce mitochondrial membrane damage. However, nutrient starvation can scavenge ROS by activating

superoxide dismutase 2 (SOD2). Therefore more MTs are available for Zn^{2+} binding, resulting in the decreased intracellular labile Zn^{2+} .⁵⁹” (See Page 21 in the revised manuscript)

Reviewer #3:

C&Q 13: He, Diao, Chen and co-authors reported a new method to simultaneously tracking Zn^{2+} in living cells based on the super-resolution morphology correlated organelles. It looks very interesting, especially with very fantastic colorful imaging with high-resolution in living cells. However, in my opinion, this is a kind of “old wine” in a new concept created by themselves. First, tracking dynamic Zn^{2+} ion in living cells is importance in understanding Zn hemostasis in bioinorganic chemistry, so development of Zn sensor is consequently important. There are many Zn fluorescence sensors reported in recent years and most of them are based on ICT mechanism. The so-called simultaneous Zn^{2+} tracking in multiple organelles is a kind of Zn sensor essentially with non-specific organelle targeting. In this aspect, the design of NapBu-BPEA is not novel.

R&A 13: Thanks for the evaluation and comment. To meet the requirement of Zn-STIMO via SIM imaging, we selected naphthalimide-derived Zn^{2+} probes due to their medium fluorescence quantum yield and fine stability, and fine Zn^{2+} sensing ability. However, the formed naphthalimide-derived Zn^{2+} probes such as Naph-BPEA and NaphEt-BPEA showing a Zn^{2+} binding-induced uniform redistribution inside cells, which disfavors Zn-STIMO. Through regulating the lipophilicity of the probes, we made NapBu-BPEA be able to distribute in multiple organelles even upon Zn^{2+} binding, favoring Zn-STIMO. This property is quite essential for this new method for labile Zn^{2+} tracking. In this respect, the design of probe is essential for the proposed Zn-STIMO.

C&Q 14: Second, the proof-of-concept experiments have been done mostly using CCCP inducing Zn^{2+} release in mitochondria. However, CCCP treatment is very complicate, as the authors also indicated. At current stage, it is lack of evidence to demonstrate the green fluorescence species is Zn NapBu-BPEA in living cells, where I suggest to perform X-ray fluorescence experiments. This is important for detecting intracellular Zn^{2+} . Many groups reported good Zn sensors in live cells have been eventually proven to be “turn on” by other species than Zn^{2+} . This should be careful.

R&A 14: Thanks for the careful evaluation and efforts. X-ray fluorescence (XRF) imaging is indeed a very effective method for detection of Zn^{2+} in cells (C. J. Fahrni, *Curr. Opin. Chem. Biol.* **2007**, *11*, 121; M. J. Pushie et al., *Chem. Rev.* **2014**, *114*, 8499). However, XRF imaging has no ability to discriminate labile Zn^{2+} from bound Zn^{2+} , and therefore offers the information of total zinc distribution in cells. This is different from the concept of Zn-STIMO to track labile Zn^{2+} dynamically in multiple organelles. Moreover, XRF experiments are performed normally with dehydrated cells, which is difficult to monitor

the dynamic change of intracellular labile Zn^{2+} in real time. We have also mentioned this in the introduction.

In order to confirm that labile Zn^{2+} enhancement is responsible for the CCCP-induced green fluorescence enhancement, the subsequent treatment with cell membrane permeable Zn^{2+} scavenger TPEN was performed, which led to the CCCP-enhanced fluorescence decreasing obviously (Figure 5). This indicated that the CCCP-enhanced signal was ascribed to labile Zn^{2+} increment. In addition, we performed the new dynamic tracking of labile Zn^{2+} in real time during autophagy (new Figure S15 in the revised SI). This experiment gave a fluorescence temporal profile which disclosed a CCCP-induced dynamic fluorescence enhancement and the subsequent Zn^{2+} scavenging-triggered fluorescence drop. This has also been discussed and highlighted in the revised manuscript (See page 14 in the revised version).

Figure S15. (a) Time-lapse SIM images of the NapBu-BPEA-stained HeLa cells recorded during incubation with 20 μ M CCCP. TPEN (100 μ M) treatment was administrated after 14 mins of CCCP treatment, and the corresponding temporal profiles of mean fluorescence intensity inside cells (b) and inside dot-shaped organelles (c).

C&Q 15: Third, tracking dynamic Zn^{2+} in living cells is very attractive. However, I did not read this interesting part about monitoring dynamic process. If there is no such study, I think this is a paper about sensor design to catch Zn^{2+} using an advances fluorescence microscopy.

R&A 15: Thanks for the constructive comment. We agree with the reviewer that dynamic tracking of labile Zn^{2+} change in intracellular multi-organelles is essential for Zn-STIMO. Then we performed the dynamic labile Zn^{2+} tracking in cells undergoing mitophagy upon CCCP treatment (Figure 7), and the results has been discussed as the follow and highlighted in the revised manuscript: **Since NapBu-BPEA can be used to monitor Zn^{2+} in different organelles simultaneously via Zn-STIMO. However, the Zn^{2+} level in cells changes from time to time, so tracking dynamic Zn^{2+} in living cells is very attractive. In view of the previous solid experimental results, we also performed dynamic labile Zn^{2+} tracking in the CCCP-induced mitophagy of HeLa cells via Zn-STIMO (Figure 7). The temporal profiles of mean fluorescence in the mitochondria, ER, and autophagosome/autolysosome (Aps/AIs) revealed that the distinct labile Zn^{2+} enhancement appeared 5 mins later than CCCP incubation, and the Aps/AIs displayed the more distinct enhancement of labile Zn^{2+} than the mitochondria and ERs (Figure 7e). This dynamic tracking of labile Zn^{2+} exactly showed the capability of Zn-STIMO via SIM imaging. Please see also Page 17 in the revised manuscript.**

Figure 7. NapBu-BPEA enables dynamically track of labile Zn^{2+} in cells undergoing mitophagy via

Zn-STIMO. (a) Time-lapse SIM images of HeLa cells stained by **NapBu-BPEA** recorded upon incubation with 20 μ M CCCP; and zoom-in images of regions of interest marked with squares (b, mitochondria), circles (c, ERs), and rounded squares (d, autophagosomes/autolysosomes). (e) Temporal profiles of mean fluorescence intensity detected for mitochondria, ERs, and autophagosomes/autolysosomes in HeLa cells showed in (a-d).

Suggestions:

C&Q 16: Pls. give a general scheme or figure to demonstrate the concept of Zn-STIMO, which is important to clarify.

R&A 16: We really appreciated this suggestion. We have given a general schematic illustration of Zn-STIMO. Please see new Figure 1a in the revised manuscript.

Figure 1a. General scheme to demonstrate Zn-STIMO.

C&Q 17: Pls. reorganize the manuscript and reduce the tedious part such as the design of probe (page 7).

R&A 17: According to the reviewer's suggestion, we have simplified the part of probe design, please see the highlighted part in pages 5-6.

C&Q 18: Pls. delete interpretation or explanation part you do not have strong evidences to support.

R&A 18: We have deleted some unfounded explanations in the main text. For examples, detailed discussion of $\log P$ and partial explanation of theoretical calculation.

REVIEWERS' COMMENTS

Reviewer #1 (Remarks to the Author):

The revised manuscript reasonably addressed the majority of the concerns. However, there is still one need to be clarified before publication.

For C&Q 3, if we need extra commercial dyes to confirm the identity of different vesicle-like structures, the title of the manuscript, i.e. "morphology-correlated organelle identification", is obviously over-claim and should be re-worded to avoid misleading statements.

Moreover, it is really not a desired property of targeting to multiple organelles, e.g., ER, lysosome, and mitochondria. In another word, if this probe locate onto multiple vesicle-like organelles, how could one discriminate them? Try various marker to discriminate them? It is hard to do high-quality labeling of multiple organelles in the same cell. Please add some discussion at least.

Reviewer #2 (Remarks to the Author):

The authors have addressed the issues of the original version.
The quality of the manuscript has been improved.

Reviewer #3 (Remarks to the Author):

I am satisfied with this revision and reply for this work. I also suggest to cite work about some luminescent Zinc complexes which are important to understand the luminescence mechanism (Chem. Commun., 2016, 52, 11583-11586; Chem. Sci., 2015, 6, 2389-2397; Chem. Sci., 2015, 6, 2389-2397; Chem. Sci., 2012, 3, 3315-3320; Chem. Commun., 2011, 47, 2435-2437).

Reviewer #2:

The authors have addressed the issues of the original version.

The quality of the manuscript has been improved.

R&A: We thank the reviewer for the positive and encouraging notes on our efforts to improve the manuscript.

Reviewer #3:

I am satisfied with this revision and reply for this work. I also suggest to cite work about some luminescent Zinc complexes which are important to understand the luminescence mechanism (Chem. Commun., 2016, 52, 11583-11586; Chem. Sci., 2015, 6, 2389-2397; Chem. Sci., 2015, 6, 2389-2397; Chem. Sci., 2012, 3, 3315-3320; Chem. Commun., 2011, 47, 2435-2437).

R&A: We thank the reviewer for the valuable suggestion on our manuscript. We have cited these papers to improve our understanding of the luminescence mechanism of Zinc complexes.